



# Oxygen isotope exchange between water and carbon dioxide in soils is controlled by pH, nitrate availability and microbial biomass through links to carbonic anhydrase activity

Sam P. Jones[1,2], Aurore Kaisermann[1], Jerome Ogee[1], Steven Wohl[1], Alexander W. Cheesman[3], Lucas A. Cernusak[3], and Lisa Wingate[1]

[1] INRA, UMR ISPA, 33140, Villenave d'Ornon, France
[2] Instituto Nacional de Pesquisas da Amazônia, Manaus – AM, CEP 69060-001, Brasil
[3] College of Science and Engineering, James Cook University, Cairns, Queensland, Australia

*Correspondence to*: Sam P. Jones (sam.p.jones@hotmail.co.uk)

**Abstract.** The oxygen isotope composition ($\delta^{18}O$) of atmospheric carbon dioxide ($CO_2$) can be used to estimate gross primary production at the ecosystem-scale and above. Understanding how and why the rate of oxygen isotope exchange between soil water and $CO_2$ ($k_{iso}$) varies can help to reduce uncertainty in the retrieval of such estimates. The expression and activity of carbonic anhydrases in soils are important drivers of variations in $k_{iso}$. Here we estimate $k_{iso}$ and measure associated soil properties in laboratory incubation experiments using 44 soils sampled from sites across western Eurasia and northeastern Australia. Observed $k_{iso}$ exceeded theoretical uncatalysed rates indicating the significant influence of carbonic anhydrases on the variability observed among the soils studied. We identify soil pH as the principal source of variation, with greater $k_{iso}$ under alkaline conditions suggesting that shifts in microbial community composition or intra-extra cellular dissolved inorganic carbon gradients induce the expression of more or higher activity forms of carbonic anhydrases. We also show for the first time in soils that the presence of nitrate under acidic conditions reduces $k_{iso}$, potentially reflecting the direct or indirect inhibition of carbonic anhydrases. This effect was confirmed by a supplementary ammonium nitrate fertilisation experiment conducted on a subset of the soils. Future changes in atmospheric nitrogen deposition or land-use may thus influence carbonic anhydrase activity. Greater microbial biomass also increased $k_{iso}$ under a given set of chemical conditions likely highlighting the ubiquity of carbonic anhydrase expression by soil microbial communities. These data provide the most extensive analysis of spatial variations in soil $k_{iso}$ to date and indicate key controls required to predict variations in $k_{iso}$ at the scales needed to improve efforts to constrain gross primary productivity using the $\delta^{18}O$ of atmospheric $CO_2$.



## 1 Introduction

The rate of oxygen isotope exchange ($k_{iso}$) between soil water and carbon dioxide ($CO_2$) is a key uncertainty in estimating the gross contribution of terrestrial uptake and release to net land-atmosphere carbon exchange from the oxygen isotope composition ($\delta^{18}O$) of atmospheric $CO_2$ (Wingate et al., 2009; Welp et al. 2011). The $\delta^{18}O$ of atmospheric $CO_2$ can be used to trace these large and opposing fluxes because the $\delta^{18}O$ of leaf-atmosphere $CO_2$ exchange tends to be enriched and distinct compared to well mixed atmospheric $CO_2$ and thus has the potential to serve as an independent tracer of gross primary production (Francey & Tans, 1987). This is the case because the leaves of plants contain considerable concentrations of carbonic anhydrases, which catalyse the hydration of aqueous $CO_2$ and in turn the exchange of oxygen isotopes with water molecules, causing $CO_2$ that interacts with a leaf but is not fixed to inherit the isotopic composition of the leaf water pool (Gillon & Yakir, 2001). As leaf water pools are small and undergo considerable enrichment during evaporation the $\delta^{18}O$ of this water is enriched relative to that of the soil water and thus $CO_2$ that has interacted with leaves has a distinct isotopic signature in the atmosphere (Francey & Tans, 1987). However, the presence of carbonic anhydrases is not limited to leaves with a number of forms also found in soils (Meredith et al., 2019), but the abundance and activity of these enzymes is poorly known and the degree to which their influence on $k_{iso}$ alters the $\delta^{18}O$ of atmospheric $CO_2$ is not well constrained. As such, improved understanding of variations in soil $k_{iso}$ benefits efforts to constrain variability and controls on carbon exchange at the ecosystem-level and above.

Comprising at least six distinct families, carbonic anhydrases have independently evolved in all domains of life in order to catalyse the reversible hydration of carbon dioxide ($CO_2$) to bicarbonate (Del Prete et al., 2014). Whilst this reaction occurs abiotically, the need for carbonic anhydrases stems from the fact that enhanced rates of hydration, $k_h$ ($s^{-1}$), are required to control the transport and availability of $CO_2$, bicarbonate and protons in numerous metabolic processes (Smith & Ferry, 2000). Unsurprisingly given their apparent ubiquity, evidence of carbonic anhydrase activity in soils indicates the expression of these enzymes directly supports the viability of microbial communities and thus plays a role in the wider biogeochemical function of the soil environment (Li et al., 2005). An indirect consequence of increased rate of hydration, $k_h$, associated with the presence of these enzymes is that it also influences the oxygen isotope composition ($\delta^{18}O$) of atmospheric $CO_2$. The $\delta^{18}O$ of atmospheric $CO_2$ is influenced by carbonic anhydrases because oxygen isotopes are exchanged between water and $CO_2$ through the reverse dehydration reaction (Mills & Urey, 1940). In a closed system at chemical equilibrium, $CO_2$ will reach isotopic equilibrium with water after some time depending on the rate, $k_{iso}$ ($s^{-1}$), of oxygen isotope exchange (Uchikawa & Zeebe, 2012). In soils the greater abundance of water molecules means that endogeneous $CO_2$ or atmospheric $CO_2$ that abiotically invades the soil tends to inherit the $\delta^{18}O$ of the soil water (Tans, 1998). The degree to which the $\delta^{18}O$ of $CO_2$ reflects a given soil water pool is determined by the residence time of dissolved $CO_2$ and the apparent $k_{iso}$ (Miller et al.,



1999). Longer residence times or greater $k_{iso}$ move the system closer to isotopic equilibrium. Resulting from the interconversion of aqueous $CO_2$ and bicarbonate, $k_{iso}$ is expected to vary as a function of the combined rates of $CO_2$

hydration, $k_h$, and hydroxylation reactions and the pH dependent speciation of dissolved inorganic carbon (Uchikawa & Zeebe, 2012). Under acidic and neutral conditions interconversion is dominated by hydration, whilst the hydroxylation becomes important under alkaline conditions as the concentration of hydroxyl anions increases (Figure 1 a). The presence of carbonic anhydrases increases the rate of the hydration reaction and the overall rate of interconversion. However, the influence of carbonic anhydrases, for a given concentration and efficiency, is also limited by the presence of high proton

concentrations under acidic conditions that inhibit de-protonation required for enzyme regeneration (Rowlett et al., 2002; Sauze et al., 2018). The $k_{iso}$ resulting from these reactions is dependent on the relative abundance of $CO_2$, which is only the dominant form of dissolved organic carbon under acidic conditions, to true carbonic acid, bicarbonate and carbonate in the system (Figure 1 b). In alkaline conditions, the predominance of bicarbonate and carbonate acts to inhibit the rate of $k_{iso}$ associated with the hydration reaction and limit the influence of hydroxylation (Figure 1 c). The fact that $k_{iso}$ inferred from

patterns in $\delta^{18}O$ of $CO_2$ fluxes observed under field (Seibt et al., 2006; Wingate et al., 2008, 2009, 2010) and laboratory conditions (Jones et al., 2017; Meredith et al., 2019; Sauze et al., 2017, 2018) can exceed uncatalysed rates predicted by this theoretical basis by up to three orders of magnitude indicates a particular need to better understand variations in the expression of carbonic anhydrases and the controls on their activity in soil environments.

The ubiquity of carbonic anhydrases expression by organisms (Smith & Ferry, 2000) has been invoked to explain variations in the rate, $k_h$, of hydration (Li et al., 2005) and the rate, $k_{iso}$, of oxygen isotope exchange (Wingate et al., 2009) in soils as a function of the size and composition of the microbial communities present. That is to say if carbonic anhydrase expression is widespread within a microbial community or a specific functional group, increases in their abundance will result in greater concentrations of carbonic anhydrases within the soil and thus greater reaction rates. Indeed, such a driver for $k_{iso}$ is

supported by experimental observations showing that rates increase with algae (Sauze et al., 2017) and α- and β-carbonic anhydrase (Meredith et al., 2019) abundances. Whilst the sensitivity of soil $k_{iso}$ to the presence of specific functional groups, like phototrophs which employ carbonic anhydrases in their carbon concentration mechanisms (Badger, 2003), is clear from these studies, the influence of wider community size and composition and it's utility in predicting variations remains uncertain. Soil $k_{iso}$ has also been shown to vary in response to soil pH, with greater rates under alkaline conditions (Sauze et

al., 2018). The observed increase, contrary to the decrease (Figure 1 c) that would be expected to be induced by the shift in dissolved inorganic carbon speciation (Figure 1 b) at a constant carbonic anhydrase concentration and efficiency (Figure 1 a), suggests a greater abundance or the presence of more efficient forms of carbonic anhydrases at higher soil pH. Such an observation may result from changes in size or composition of the microbial communities involved as discussed (Sauze et al., 2017, 2018). Alternatively this pattern might be driven by the up-regulation of carbonic anhydrase expression by

organisms, which tend to maintain more neutral conditions than their environment (Krulwich et al., 2011), in order to control





intra-extra cellular dissolved inorganic carbon gradients (Smith & Ferry, 2000) in response to changes in extra-cellular $CO_2$ and bicarbonate availability (Figure 1 b). Such a control may be supported by the fact both bacteria and fungi grown under $CO_2$ limited conditions have been shown to increase the expression of carbonic anhydrases (Kaur et al., 2009; Kozliak et al., 1995; Merlin et al., 2003). Indeed, such a response in order to maintain the supply of $CO_2$ in bicarbonate dominated systems

has been documented for aquatic phototrophs (Hopkinson et al., 2013). The chemistry of non-carbon anions may also play a role in controlling the activity of carbonic anhydrases (Tibell et al., 1984) with the presence of phosphate for example potentially inhibiting extra-cellular carbonic anhydrase activity in soil solutions and thus decreasing $k_{iso}$ (Sauze et al., 2018). In this respect, the fact that nitrate ($NO_3^-$) has been shown to inhibit carbonic anhydrases (Peltier et al., 1995) suggests that the inorganic nitrogen chemistry of soil solutions may exert a control, particularly in the context of fertilised agriculture soils

or increased atmospheric nitrogen deposition. Indeed, the inhibition of soil carbonic anhyrases by nitrogen fertilisation has been inferred from measurements of carbonyl sulphide exchange (Kaisermann et al., 2018b), but the influence on $k_{iso}$ has yet to be considered.

Here we investigate variations in the rate of oxygen isotope exchange, $k_{iso}$, through controlled laboratory gas exchange

measurements on soil incubations. To understand the drivers of these variations we measured soils, with different chemical and physical properties, sampled from 44 sites across western Eurasia and northeastern Australia. We also conducted a fertilisation experiment on a subset of these soils to investigate the influence of changes in nitrogen availability. We tested three specific, non-exclusive, hypotheses; 1) $k_{iso}$ increases with increases in microbial biomass reflecting the common nature of carbonic anhydrase expression by soil organisms (H1), 2) $k_{iso}$ increases with increases in soil pH reflecting an increase in

the amount or efficiency of expressed carbonic anhydrases either because of the response of organisms to unfavourable gradients in intra-extra cellular dissolved inorganic carbon under alkaline conditions or a shift in active functional groups (H2), and 3) $k_{iso}$ will decrease with increases in $NO_3^-$ availability as it binds with carbonic anhydrases and directly inhibits enzymatic activity (H3). For the Eurasian soils we also compare these drivers to the predictive power of relatively invariant soil properties that might be used to estimate the rate of exchange in soils at the regional scale and above as required by

efforts to better constrain gross primary production.

## 2 Methods

To investigate the outlined hypotheses two similar measurement campaigns, each consisting of a spatial survey and an ammonium nitrate ($NH_4NO_3$) addition experiment, were conducted aimed at characterising the controls on variations in the rate of oxygen isotope exchange, $k_{iso}$, across soils from a wide range of environments. In both cases we estimated $k_{iso}$ from

gas exchange and soil physical property measurements (Jones et al., 2017; Sauze et al., 2018) and subsequently measured the pH, microbial biomass, $NO_3^-$ availability and $NH_4^+$ availability of the incubated soils to investigate the controls on this



activity. The first campaign focused on soils sampled from across western Eurasia (EUR) whilst the second (AUS) focused on soils sampled in north Queensland, Australia. Sampling sites were broadly classified based on the principle land-cover reported by previous studies or observed during sampling and climatic zone as indicted by the Köppen-Geiger climate
classification map of Kottek et al., (2006) and Rubel et al., (2017).

### 2.1 Soil sampling and incubation preparation

For the EUR campaign, collaborators (see acknowledgements) sampled the superficial 10 cm of soil at three locations within 27 sites during the Northern hemisphere summer of 2016 and shipped the soil samples to the Bordeaux-Aquitaine Center of
the National Institute of Agricultural Research, France (Figure S1 a). These sites fell within Subarctic (Dfc; n= 6 ), Temperate oceanic (Cfb; n= 13), Hot-summer Mediterranean (Csa; n = 7) and Hot semi-arid (Bsh; n = 1) climate zones and were principally found in forests (n = 16) and grasslands (n = 6). The other remaining sites were located in an agricultural field (n = 1), a peatland (n = 1) and orchards (n = 3). Upon receipt, samples were passed through a 4 mm sieve and mixed to create one homogeneous sample for each site. These soils were stored at 4 °C. A sub-sample of each of these soils was used
to determine the initial water content and the soil water holding capacity (Haney & Haney, 2010). For each soil three replicate incubations were prepared with glass jars of 15.54 cm in height and an internal diameter of 8.74 cm. A jar was filled with the wet weight equivalent of 115 to 300 g of dry soil and the water content adjusted to 30 % of the water holding capacity to create a soil column with a surface area of 60.0 $cm^2$ and a depth of approximately 4 to 7 cm. The jar was then pre-incubated in a climate-controlled cabinet (MD1400, Snijders, Tillburg, NL) for two weeks in the dark at 22 ± 1 °C. This
cabinet was continuously flushed with approximately 20 L $min^{-1}$ of  ambient air provided by a pump with an inlet outside of the building to avoid exposing the soil to elevated $CO_2$ concentrations found within the laboratory. During this period, soil water content was periodically adjusted to account for evaporation. Approximately 18 hours prior to measurement the jar was closed with a screw-tight glass lid equipped with inlet and outlet connections and flushed at 250 mL $min^{-1}$ with dry, synthetic air to promote steady-state conditions. This flow was produced using an in-house dilution system that mixed pure
$CO_2$ from a cylinder into $CO_2$-free air generated by an air compressor (FM2 Atlas Copto, Nacka, Sweden) equipped with a scrubbing column (Ecodry K-MT6, Parker Hannifin, USA). This system was set to achieve a $CO_2$ concentration of 400 ± 5 ppm and, reflecting the origin of the $CO_2$ in the cylinder used, had a $\delta^{18}O$ of approximately −25 ‰ $VPDB_g$. Subsequently the jar was removed to conduct gas exchange and soil property measurements.

For the AUS campaign, we sampled the superficial 10 cm of soil at four locations within 17 sites during July of 2017 and returned these samples to the Cairns campus of James Cook University (Figure S1 b). These sites fell within Tropical monsoon (Am; n = 3), Humid subtropical (Cfa; n = 9) and Monsoon-influenced humid subtropical (Cwa; n= 5) climate zones and were principally found in forests (n = 9) and savannas (n = 6), with the other remaining sites located in a pasture





(n = 1) and a stunted shrub-rich forest (n = 1). These soils were passed through a 4 mm sieve and mixed to create a
homogenous sample for each site. A sub-sample of each of these soils was used to determine the initial water content and
estimate the re-packed bulk density of the soils. As with the EUR campaign, three replicate incubations were prepared in
glass jars for each soil. These jars had a height of 11.56 cm and an internal diameter of 7.45 cm. A jar was filled with the wet
weight equivalent of 215 to 450 g of dry soil and the water content adjusted to 30 % water-filled pore space to create a soil
column with a surface area of 43.5 cm$^2$ and a depth of approximately 8.5 cm. The jar was then pre-incubated in an insulated
box for one week in the dark at 23 ± 1 °C with periodic adjustments to the water content to account for evaporation. This box
was continuously flushed with approximately 10 L min$^{-1}$ of ambient air provided by an air compressor that serviced building
wide laboratory air distribution. The concentration of $CO_2$ in this air was approximately 420 ppm and, and reflecting it's
atmospheric origin, had a $\delta^{18}O$ of approximately 0 ‰ VPDB$_g$. Following pre-incubation the jar was removed to conduct gas
exchange and soil property measurements.


An NH$_4$NO$_3$ addition experiment was conducted in both campaigns. To do so an additional three replica incubations were
prepared as described above, with these untreated soils serving as controls, for nine and five soils of the EUR and AUS
campaigns respectively. Prior to the pre-incubation step, 0.7 mg of NH$_4$NO$_3$ g dry soil$^{-1}$ was added dissolved in the water
used to adjust the water content. This quantity was chosen following Ramirez et al., (2012) to approximate a treatment
comparable to typical field studies.

## 2.1 Gas exchange measurements

Gas exchange measurements were made using a similar experimental set-up to that described  in Jones et al., (2017). Each
jar was connected to a gas delivery system that supplied one of two gas sources, $\delta_{b,atm}$ or $\delta_{b,mix}$, to its inlet. The first inlet
condition, $\delta_{b,atm}$, consisted of a continuous flow of atmospheric air pumped from an external buffer volume, through a
Drierite column (W. A. Hammond DRIERITE Co. LTD, USA) to dry the air and directly to the inlet of the jar. The second
condition, $\delta_{b,mix}$, was produced by a second continuous flow of atmospheric air pumped from the buffer, through a soda lime
column to remove $CO_2$ and a second Drierite column. A mass-flow controller was used to dilute pure $CO_2$ from a cylinder
into this dry $CO_2$ free air and then this mix was supplied to the inlet of the jar. The flow rate of pure $CO_2$ was controlled to
match the concentration of the $CO_2$ in $\delta_{b,mix}$ to that of $\delta_{b,atm}$ using a control loop feedback based on the difference in
concentration between sub-samples of both flows measured with an infra-red $CO_2$ analyser (Li-6262, LI-COR Biosciences,
USA). By doing so the principal difference between the two conditions was the isotopic composition of the $CO_2$ present
reflecting its origin in the atmosphere ($\delta^{18}O$-$CO_2$ of $\delta_{b,atm}$ = −1.41 ± 2.17 ‰ VPDB$_g$) or  a cylinder ($\delta^{18}O$-$CO_2$ of $\delta_{b,mix}$ =
−25.33 ± 0.30 ‰ VPDB$_g$). The delivery of either gas to the inlet of the jar was operated by a valve manifold and micro-



controller. Following the manifold, the selected gas stream was split into a chamber line, to which the jar was connected, and a bypass line that terminated at open splits in front of a valve connected to the sample inlet of a $CO_2$ isotope ratio infrared spectrometer (Delta Ray IRIS, Thermo Fischer Scientific, Germany). The flow rate of the chamber line was limited to 171.48 μmol s$^{-1}$ using a mass-flow controller. The micro-controller was set to supply first one inlet condition through the manifold to the chamber and bypass line and then switch to the second inlet condition. Both inlet conditions were supplied

for either 32 (EUR) or 34 (AUS) minutes. The first 20 (EUR) or 22 (AUS) minutes under each condition were used to flush the system and promote steady-state conditions in the incubation jar. After this period, the final 12 minutes during which the condition was supplied before switching was used for gas-exchange measurements. During this 12 minute measurement period the valve in front of the IRIS switched three times between the chamber and bypass line at two minute intervals. Calibration gas was measured every 16 (EUR) or 18 (AUS) minutes with sequential two minute measurements of two

cylinders containing synthetic air with different $CO_2$ concentrations but similar isotopic compositions. The concentrations of $^{12}C^{16}O^{16}O$, $^{13}C^{16}O^{16}O$ and $^{12}C^{18}O^{16}O$ recorded by the IRIS were processed as described in detail by Jones et al. (2017) to average the final 40 s of data collected for each two minute interval discussed and calculate corrected concentrations and isotope ratios.

Reflecting the pre-incubation conditions, measurements for EUR began with $\delta_{b,mix}$ as the inlet condition before switching to

$\delta_{b,atm}$, whilst for AUS the sequence began with $\delta_{b,atm}$ and then switched to $\delta_{b,mix}$. For EUR, the calibration cylinders (21 % $O_2$ and 0.93 % Ar in a $N_2$ balance, Deuste Steinger GmbH, Germany) had total concentration, carbon isotope composition and $\delta^{18}O$ of $CO_2$, respectively, of 380.26 ppm , −3.06 ‰ VPDB, and −14.63 ‰ VPDB$_g$ for the first cylinder, and 481.62 ppm , −3.07 ‰ VPDB and 14.70 ‰ VPDB$_g$ for the second cylinder (IsoLab, Max Planck Institute for Biogeochemistry, Germany). For AUS, the calibration cylinders (21 % $O_2$ and 1.12 % Ar in a $N_2$ balance, BOC, Australia) had total concentration, carbon

isotope composition and $\delta^{18}O$ of $CO_2$, respectively, of 386.7 ppm, −33.42 ‰ VPDB and −26.33 ‰ VPDB$_g$, for the first cylinder, and 486.7 ppm, −33.64 ‰ VPDB and −26.60 ‰ VPDB$_g$ for the second cylinder (Farquhar Laboratory, Australian National University, Australia).

The net $CO_2$ flux, $F_R$ (μmol m$^{-2}$ s$^{-1}$), was calculated from corrected values for the three pairs of chamber and bypass line measurements made at each inlet condition following Eq. (1):

$$F_R = \frac{u}{A}\left(C_c - C_b\right)$$

210 , (1)

where u is the flow rate (mol m$^{-3}$ s$^{-1}$) through the chamber line, $C_c$ is the total $CO_2$ concentration (ppm) of the chamber line, $C_b$ is the total $CO_2$ concentration (ppm) of the bypass line and A is the surface area (m$^2$) of the soil in the chamber. The resultant three values for each inlet condition were then averaged to yield a single flux rate. Similarly the $\delta^{18}O$ of $CO_2$ exchange, $\delta_R$ (‰ VPDB$_g$), was calculated following Eq. (2):

$$\delta_R = \frac{\left(\delta_c C_{c,12} - \delta_b C_{b,12}\right)}{\left(C_{c,12} - C_{b,12}\right)} \, ,$$
(2)

where $\delta_c$ is the $\delta^{18}O$ of $CO_2$ (‰ VPDB$_g$) in the chamber line, $\delta_b$ is the $\delta^{18}O$ of $CO_2$ (‰ VPDB$_g$) in the bypass line, $C_{c,12}$ (ppm) is the concentration of $^{12}C^{16}O^{16}O$ in the chamber line and $C_{b,12}$ (ppm) is the concentration of $^{12}C^{16}O^{16}O$ in the bypass line.

**2.3 Soil properties**

After being disconnected from the gas exchange system, a jar was weighed to determine the wet weight of the incubated soil
and the total soil depth, $z_{max}$ (m), measured using a caliper. Soil was then removed from the jar to determine soil water content, pH, microbial biomass, $NO_3^-$ availability and $NH_4^+$ availability. Soil water contents were determined gravimetrically for sub-samples based on water loss after oven drying for 24 hours at 105 °C. In the EUR campaign, soil water content was determined for three, 1.5 cm thick intervals between 0.0 and 4.5 cm depth. An average gravimetric water content (g g dry soil$^{-1}$) was calculated for the soil column after weighting by total soil depth. In the AUS campaign, soil water content was
determined for a single sample covering the total soil depth. Soil bulk density (g cm$^{-3}$) was calculated from the gravimetric water content, the wet weight of the soil in the jar and the volume of the soil column. Total porosity, $\phi_t$, was calculated from bulk density assuming a particle density of 2.65 g cm$^{-3}$ (Linn & Doran, 1984). Volumetric water content, $\theta_w$ (m$^3$ m$^{-3}$), was calculated as the product of gravimetric water content and bulk density. The soil air-filled porosity, $\phi_a$, was calculated as the difference between the total porosity and volumetric water content. The remaining soil column in the jar was then mixed and
sub-samples were taken to determine pH, microbial biomass, $NO_3^-$ availability and $NH_4^+$ availability. Soil pH was determined in a slurry with a dry weight equivalent soil-to-water ratio of 1:5. Soil microbial biomass (μg C g dry soil$^{-1}$) was determined based on the difference between dissolved carbon extracted from non-fumigated and chloroform-fumigated sub-samples using a slurry with a dry weight equivalent soil-to-potassium sulphate solution (0.5 M) ratio of 1:5 and an extraction efficiency value of 0.35. Available $NO_3^-$ (μg N g dry soil$^{-1}$) and $NH_4^+$ (μg N g dry soil$^{-1}$) were extracted in a slurry with a dry
weight equivalent soil-to-potassium chloride solution (1 M) ratio of 1:5. These extracts were filtered, frozen at −20 °C and shipped on dry ice to commercial laboratories (EUR: LAS INRA Hauts-de-France, Arras, France; AUS: ASL Environmental, Brisbane, Queensland, Australia) for determination of dissolved carbon, $NO_3^-$ and $NH_4^+$ concentrations. Sub-samples of the homogenised soil used to fill jars in the EUR campaign were also taken to determine soil texture and carbon, nitrogen and carbonate content by sampling site as part of a related study (Kaisermann et al., 2018a).






## 2.4 Estimating the oxygen isotope exchange rate

Following Jones et al., (2017) the rate of oxygen isotope exchange between soil water and $CO_2$, $k_{iso}$, was estimated from the inverse of the slope of the linear relationship between the $\delta^{18}O$ of $CO_2$ exchange and the $\delta^{18}O$ of $CO_2$ at the soil surface. Briefly, under the two gas-exchange measurement conditions induced by varying the $\delta^{18}O$ of $CO_2$ at the incubation inlet

($\delta_{b,mix}$ and $\delta_{b,atm}$), the invasion flux or piston velocity of $CO_2$, $v_{inv}$ (m s$^{-1}$), can be estimated following Eq. (3):

$$v_{inv} = \frac{F_{R,\mu}}{C_{a,\mu}} \frac{\left(\delta_{R,mix} - \delta_{R,atm}\right)}{\left(\delta_{a,atm} - \delta_{a,mix}\right)} \quad, \tag{3}$$

where $\delta_R$ (‰ VPDB$_g$) is the $\delta^{18}O$ of $CO_2$ exchange and $\delta_a$ (‰ VPDB$_g$) is the $\delta^{18}O$ of $CO_2$ at the soil surface under the two different inlet conditions ($\delta_{b,mix}$ and $\delta_{b,atm}$) and $F_{R,\mu}$ (µmol m$^{-2}$ s$^{-1}$) is the mean net $CO_2$ flux under both conditions and $C_{a,\mu}$ (µmol m$^{-3}$) is the mean total $CO_2$ concentration at the soil surface measured under both conditions. Both $\delta_a$ and $C_a$ were

assumed equal to the $\delta_c$ and $C_c$ measured in the chamber line as discussed previously. To correct for the influence of boundary conditions found at the bottom of incubation jars, particularly in shallower soil columns, the soil-depth adjusted invasion flux, $\tilde{v}_{inv}$ (m s$^{-1}$), was determined iteratively to satisfy Eq. (4):

$$0 = \tilde{v}_{inv} \tanh\left(\frac{\tilde{v}_{inv} z_{max}}{\kappa \phi_a D}\right) - v_{inv} \quad, \tag{4}$$

where $z_{max}$ (m) is the total soil-column depth, $\kappa$ is soil tortuosity calculated here following the formulation of Moldrup et al.

(2003) for repacked soils, D (m$^2$ s$^{-1}$) is the diffusivity of $^{12}C^{16}O^{18}O$ in air (Massman, 1998; Tans, 1998) and $\phi_a$ is the air-filled porosity of the soil (see Sauze et al., (2018) for the derivation). Subsequently $k_{iso}$ (s$^{-1}$) was calculated following Eq. (5):

$$k_{iso} = \frac{\tilde{v}_{inv}^2}{\kappa \phi_a DB\theta_w} \quad, \tag{5}$$

where B (m$^3$ m$^{-3}$) is the Bunsen solubility coefficient for $CO_2$ in water (Weiss, 1974) and $\theta_w$ (m$^3$ m$^{-3}$) is the soil volumetric water content.

## 2.5 Statistical analyses

Statistical analyses were conducted in R version 3.5 (R Core Team, 2019). Of the 174 individual incubations prepared, 10 were excluded from the dataset because a record for one of the variables of interest; the rate, $k_{iso}$, of oxygen isotope exchange, pH, microbial biomass, $NO_3^-$ availability or $NH_4^+$ availability, was missing. For the remaining 164 incubations



with complete records, these variables were averaged by sampling site and, for the relevant subset, by whether they received
a $NH_4NO_3$ addition.

The resultant dataset consisted of mean observations for 44 untreated soils (n = 27 / EUR and 17 / AUS) and 14 soils (n = 9 /
EUR and 5 / AUS) that received a $NH_4NO_3$ addition. Spatial controls on $k_{iso}$ were investigated across the means of untreated
soils. Correlations between $k_{iso}$, pH, microbial biomass, $NO_3^-$ availability and $NH_4^+$ availability were investigated through the
Spearman's rank correlation between pairs of variables. To test the outlined hypotheses, a multiple generalised linear
modelling approach was used to investigate which variables best explained variations in $k_{iso}$ (Thomas et al., 2017). As pH
and $NH_4^+$ availability were strongly negatively correlated (Spearman's $\rho = -0.73$), presumably reflecting the pH dependency
of $NH_4^+$ and ammonia speciation, these were not considered together in the same model whilst all other possible
combinations, including sampling campaign (EUR or AUS) to test for the undue influence of systematic experimental
differences, were tested. Combinations were limited to models containing four or less predictive terms to prevent over-fitting
and each independent variable was centered and scaled to facilitate comparison among the different measurement scales. The
model structure and predictive terms included in the minimal adequate model required to explain variations in $k_{iso}$ were
selected based on comparison of sample size corrected Aikake's Information Criterion (AICc) and visual assessment of the
conformity of model residuals to the assumptions of normality, homogeneity and the absence of unduly influential
observations. This model was subsequently re-fitted with the original unstandardised variables. The same approach, limited
to two-term models, was also applied to only the 27 soils from the EUR sampling campaign and extended to consider the
relationships with soil texture and carbon and nitrogen contents to investigate their utility in upscaling efforts.

To investigate the influence of the $NH_4NO_3$ addition on the rate of oxygen isotope exchange, $k_{iso}$, the variables of interest
were expressed as the ratio of the mean of the soils that received an addition and that of their respective untreated
counterparts with quotients smaller and greater than one respectively indicating a reduction and increase following addition.
Correlations between these fractional changes for $k_{iso}$, microbial biomass, pH, $NO_3^-$ availability and $NH_4^+$ availability were
investigated through the Spearman's rank correlation between pairs of variables. The minimal adequate, generalised linear
model describing the fractional change in $k_{iso}$ across these soils was investigated by comparing the AICc and visual
inspection of the residuals for models that considered  each independent variable separately to avoid over-fitting.



## 3 Results

### 3.1 Variations among untreated soils

Clear differences in the rate, $k_{iso}$, of oxygen isotope exchange, pH, microbial biomass, $NO_3^-$ availability and $NH_4^+$ availability were not apparent as a function of sampling site climatic zone or land-cover (Figure 2). Estimates of $k_{iso}$ ranged from 0.01 to

0.40 $s^{-1}$ with the greatest rates occurring in soils sampled from hot-summer Mediterranean (Csa) , hot semi-arid (Bsh)  and subtropical (Cfa and Cwa) climates (Figure 2 a). Soil pH ranged from 3.9 to 8.6 and were mostly acidic or neutral with alkaline conditions  only found for  soils sampled from hot-summer Mediterranean (Csa) and hot semi-arid (Bsh) climates (Figure 2 b). Ranging from 98.5 to 2898.1 µg C g dry soil$^{-1}$, microbial biomass did not appear to vary systematically with sampling site origin. Available $NO_3^-$ (Figure 2 d) ranged from 0.3 to 275.7 µg N g dry soil$^{-1}$ and available $NH_4^+$ (Figure 2 e)

ranged from 2.5 to 64.7 µg N g dry soil$^{-1}$ with greatest availability found in soils sampled from temperate climates.

Individual relationships between pairs of these variables were investigated through Spearman's rank correlation (Table 1). Strong, significant correlations ($p < 0.05$) were only found between the rate of oxygen isotope exchange, $k_{iso}$, and soil pH (Spearman's $\rho = 0.58$), $k_{iso}$ and $NH_4^+$ availability (Spearman's $\rho = -0.62$), and soil pH and $NH_4^+$ availability (Spearman's $\rho =$

$-0.73$). Correlations between all other variable pairings were weaker and non-significant ($p > 0.05$).

Based on AICc and visual inspection of model fit and residuals, the structure of the generalised linear model describing variations in the rate of oxygen isotope exchange, $k_{iso}$, as the response variable was specified with a gaussian error distribution and log-link function (Thomas et al., 2017). The minimal adequate model with this structure (Figure S2)

included the additive effects of soil pH (0.122), the natural logarithm of $NO_3^-$ availability ($-0.730$) and the natural logarithm of microbial biomass (0.463), the interaction between soil pH and the natural logarithm of $NO_3^-$ availability (0.109) and an intercept term ($-6.046$). This model explained 71 % of the deviance in $k_{iso}$ (Figure 4 a) compared to the null model containing only an intercept term. The best model  had an AICc that was 6.1 lower than the next best alternative model which omitted the interaction term, 7.1 lower than the closest model containing sampling campaign and 13.3 lower than the

closest model containing the natural logarithm of $NH_4^+$ availability. The AICc values of single-term models containing only pH or the natural logarithms of microbial biomass or $NO_3^-$ availability were respectively 21.6, 43.6, and 50.2 greater than the best model. The selected model predicts the response variable, $k_{iso-pred}$ ($s^{-1}$), in the original measurement units following Eq. (6):

$$\ln\left(k_{iso-pred}\right) = 0.122 \times pH - 0.730 \times \ln\left(NO_3^-\right) + 0.463 \times \ln\left(MB\right) + 0.109 \times pH \times \ln\left(NO_3^-\right) - 6.046 \quad , (6)$$





where pH is soil pH, $NO_3^-$ is $NO_3^-$ availability (μg N g dry $soil^{-1}$) and MB is microbial biomass (μg C g dry $soil^{-1}$). The model predicts that variations in $k_{iso}$ result from positive correlations with soil pH (Figure 3 a) and microbial biomass (Figure 3 c) and negative correlation with $NO_3^-$ availability. The interaction between soil pH and $NO_3^-$ availability is such that the negative influence of $NO_3^-$ on $k_{iso}$ occurs mainly under acidic conditions and is marginal at neutral to alkaline pH (Figure 3 b).


As with the full dataset, across the 27 soils from the EUR sampling campaign the strongest relationship with the rate of oxygen isotope exchange, $k_{iso}$, was found with pH (Spearman's ρ = 0.58), whilst a weaker but still significant (p < 0.05) relationship with $NO_3^-$ availability (Spearman's ρ = −0.42) was also identified. No significant (p > 0.05) relationships between $k_{iso}$ and clay (Spearman's ρ = 0.11), silt (Spearman's ρ = −0.18), sand (Spearman's ρ = 0.00), carbon (Spearman's ρ = −0.03) or nitrogen (Spearman's ρ = −0.32) contents were found, whilst, the relationship with the ratio between total carbon and nitrogen content (Spearman's ρ = 0.38) was marginal (p = 0.05). The minimal adequate generalised linear model explaining variations in $k_{iso}$ selected from only the relatively invariant properties of soil texture and carbon and nitrogen content included only the intercept (−2.128) and the effect of nitrogen content (−0.119). This model explained 11 % of the deviance in $k_{iso}$ compared to the null model. After inclusion of soil pH, the minimal adequate model included the intercept (−4.535) and the additive effects of soil pH (0.4028) and clay content (−0.0017). This model explained 61 % of the deviance in $k_{iso}$ compared to 54 % for the model containing only the intercept (−4.535) and influence of soil pH (0.339).

### 3.2 Variations induced by NH₄NO₃ addition

The addition of $NH_4NO_3$ systematically increased available $NO_3^-$ and $NH_4^+$ and decreased the rate of oxygen isotope exchange, $k_{iso}$, and soil pH. Available $NO_3^-$ and $NH_4^+$ in the treated soils that received the $NH_4NO_3$ addition were respectively 1.9 to 173.6 and 3.7 to 18.8 times greater than in the corresponding untreated soils. Soil pH and $k_{iso}$ were respectively 0.86 to 0.98 and 0.21 to 0.76 times smaller in the soils that received the addition than in the corresponding untreated soils. The addition did not have a systematic influence on microbial biomass, which varied between 0.64 and 1.84 of the magnitude in the corresponding untreated soils.


Individual relationships between pairs of these fractional changes were investigated through Spearman's rank correlation (Table 2). Strong, significant correlations (p < 0.05) for variable pairs were found between the fractional changes in $k_{iso}$ and soil pH (Spearman's ρ = 0.57), $k_{iso}$ and $NO_3^-$ availability (Spearman's ρ = −0.84), and soil pH and $NO_3^-$ availability (Spearman's ρ = −0.75). Correlations between all other variable pairings were weaker and non-significant (p > 0.05).




Based on AICc and visual inspection of model fit and residuals, the structure of the generalised linear model describing variations in the fractional change in the rate of oxygen isotope exchange, $k_{iso}$, as the response variable was specified with a betareg error distribution and identity link function (Thomas et al., 2017). The minimal adequate, single term model with this structure included the natural logarithm of the fractional change in $NO_3^-$ availability (−0.499) and an intercept term (1.219).

This model predicts the variations in the fractional change in $k_{iso}$ following $NH_4NO_3$ addition across soils from the 14 sites considered result from a negative relationship with fractional changes in $NO_3^-$ availability (Figure 5). This relationship explained 76 % of the deviance in the fractional change in $k_{iso}$ and the model had an AICc that was 13.2 lower than the next best alternative model which included the fractional change in soil pH and an intercept term.

## 4 Discussion

This study aimed to investigate the drivers of variations in the rate of oxygen isotope exchange, $k_{iso}$, between soil water and $CO_2$ with a view to improving our ability to predict the influence of soils on the $\delta^{18}O$ of atmospheric $CO_2$ and our understanding of dynamics in the activity of carbonic anhydrases expressed by soil microbial communities. To do so, controlled incubation experiments were conducted with soils sampled from 44 sites across western Eurasia and northeastern Australia in order to estimate $k_{iso}$ and metrics relating to hypothesised controls on this activity. Estimates of $k_{iso}$ for untreated

soils ranged from 0.01 to 0.4 $s^{-1}$ (Figure 2 a). In all cases these rates exceeded theoretical uncatalysed rates of oxygen isotope exchange calculated for the incubation conditions (Uchikawa & Zeebe, 2012), which ranged from 0.00008 to 0.008 $s^{-1}$, indicating the presence of active carbonic anhydrases. These observations, with a median of 0.07 $s^{-1}$, are in good agreement with a number of previous studies which estimated $k_{iso}$ ranging from 0.03 to 0.15 $s^{-1}$ for sieved soils incubated in the dark (Jones et al., 2017; Sauze et al., 2018, 2017), but are somewhat lower than those reported by Meredith et al. (2019) with a

median and range of 0.46 $s^{-1}$ and 0.08 to 0.88 $s^{-1}$, respectively. These greater $k_{iso}$ are more comparable to those, ranging from 0.01 to 0.75 $s^{-1}$, reported by Sauze et al. (2017) for soils with well developed algal communities. Direct comparison with field observations is non-trivial because these older studies tend to address soil carbonic anhydrase activity as a range of enhancement factors over a temperature sensitive uncatalysed rate of hydration (Seibt et al., 2006; Wingate et al., 2008, 2009, 2010). However, using the mid-point of the enhancement factors and soil temperatures reported by Wingate et al.

(2009), we can estimate that $k_{iso}$ varied between 0.04 and 13 $s^{-1}$ with a median of 0.31 $s^{-1}$ across the seven ecosystems studied. Whether the potential for $k_{iso}$ to be orders of magnitude greater in the field than in incubation studies is an artefact of the sensitivity of the methodology applied to estimate the isotopic composition of the soil water pool from which exchanged $CO_2$ inherits it's signal (Jones et al., 2017) or a reduction in carbonic anhydrases following the exclusion of potentially active elements of the rhizosphere (Li et al., 2005), phototrophs (Sauze et al., 2017) or fauna in sieved soils remains an unresolved

but key question.





We hypothesised that the rate of oxygen isotope exchange, $k_{iso}$, might be positively correlated with microbial biomass (H1), positively correlated with soil pH (H2) and negatively correlated with $NO_3^-$ availability (H3). We found evidence in support of all three hypotheses with the minimal adequate statistical model explaining variations in $k_{iso}$ observed across untreated

soils including all three of these terms (Eq. 6). The model suggests that the positive relationship with soil pH (Figure 3 a), the strongest single predictor of variations in $k_{iso}$, reinforces the emergent view of soil pH as the principal driver of variations in carbonic anhydrase expression by soil microbial communities (Sauze et al., 2018). Marked increases in $k_{iso}$ under alkaline conditions likely reflects a shift in microbial community towards organisms that express more or more efficient carbonic anhydrases than those found under acidic conditions (Meredith et al., 2019; Sauze et al., 2018, 2017) and the need for

organisms to up-regulate carbonic anhydrases expression. This may be required in order to control the transport and availability of $CO_2$ and bicarbonate in response to the pH dependent speciation of dissolved inorganic carbon (Figure 1 b) as has been observed for both intra- and extra-cellular carbonic anhydrase activity in non-soil settings (Hopkinson et al., 2013; Kaur et al., 2009; Kozliak et al., 1995; Merlin et al., 2003). Similarly, in the positive relationship with microbial biomass (Figure 3 c) we find support for a secondary role for the expected link between the abundance of organisms likely to be

expressing carbonic anhydrase and $k_{iso}$ for a given set of biogeochemical conditions (Sauze et al., 2017). Finally, through the negative relationship with $NO_3^-$ availability (Figure 3 b) we show for the first time that $k_{iso}$ in soils is sensitive to dissolved inorganic nitrogen chemistry. Outwith soils, anions including $NO_3^-$ have been shown to inhibit carbonic anhydrase activity by binding with the enzyme (Peltier et al., 1995; Tibell et al., 1984). The fact that this binding and subsequent inhibition of carbonic anhydrase activity has been shown to be more efficient under acidic conditions but have minimal influence at high

pH may reflect the role of protonation in this behaviour (Johansson & Forsman, 1993, 1994). Interestingly, the interaction between soil pH and $NO_3^-$ availability identified here, leading to a larger negative influence of $NO_3^-$ availability under acidic conditions (Figure 3 b), is in agreement with this observation. This suggests that the influence of $NO_3^-$ availability on carbonic anhydrases activity is likely minimal in neutral and alkaline soils and the constraints imposed by pH and microbial community size are of greater importance. To better understand the relationship between $k_{iso}$ and soil inorganic nitrogen we

conducted an $NH_4NO_3$ addition experiment. As in other studies, the addition of $NH_4NO_3$ not only increased the availability of $NO_3^-$ and $NH_4^+$ but also acted to decrease soil pH and caused non-systematic changes in microbial biomass (Zhang et al., 2017). Reflecting the different magnitudes of these changes, the observed decrease in $k_{iso}$ in soils receiving the addition relative to their untreated counterparts was best explained by the increase in $NO_3^-$ availability (Figure 5). Notably the weak relationship between changes in $k_{iso}$ and $NH_4^+$ availability identified in this experiment (Table 2) suggests the relationship

between these variables across the untreated soils (Table 1) does indeed reflect the pH sensitivity of ammonia speciation rather than a direct causal link. The negative relationship between $NO_3^-$ availability and $k_{iso}$ appears to support the proposed mechanism of carbonic anhydrases inhibition. However, an alternative explanation, invoked to explain reductions in the activity of enzymes involved in nitrogen acquisition following fertilisation (Zhang et al., 2017), may be that carbonic





anhydrases play some role in the soil nitrogen cycle that is alleviated by increases in $NO_3^-$ availability following $NH_4NO_3$
addition and thus leads to down-regulation of expression (DiMario et al., 2017; Kalloniati et al., 2009; Rigobello-Masini et al., 2006). Indeed, such a function would help explain why the microbial communities in the untreated acidic, higher relative to lower $NO_3^-$ availability soils do not appear to need to compensate for the inhibition of carbonic anhydrases as we might expect from the economic theory of enzyme investment if they are facilitating important metabolic reactions (Burns et al., 2013). Much needed development of our understanding of the intra- and extra-cellular distribution of soil carbonic
anhydrases and their relationship to spatial and temporal variations in chemical conditions experienced by the microbial communities that express them are required to confirm the mechanistic link among these observations.

Improvements to our ability to predict the influence of soils on the the $\delta^{18}O$ of atmospheric $CO_2$ are important in refining the use of this tracer to constrain gross primary production at the ecosystem-scale and above (Wingate et al., 2009; Welp et al.,
2011) . The absence of strong patterns with climate or land-cover in this study may well reflect the fact that the temperature and moisture conditions used are unrepresentative of field conditions especially for colder and drier sites (Figure 2 a). Whether or not up-scaling based on such classes is feasible is somewhat unknown (Wingate et al., 2009). However, the data reported here does provide the basis for an empirical approach to predicting the rate of oxygen isotope exchange, $k_{iso}$, for a given soil (Figure 3). The minimal adequate statistical model described (Eq. 6) was able to provide broadly unbiased
predictions of variations observed in $k_{iso}$ across the untreated soils of the 44 sites considered (Figure 4 a). Indeed, broad agreement between predictions of the fractional changes in $k_{iso}$ between untreated and treated, which were not used in model selection, soils following the $NH_4NO_3$ addition encouragingly suggest that this model could be used to provide reasonable predictions of $k_{iso}$ for other soils (Figure 4 b). More observations from alkaline soils are required to reduce uncertainty found at greater $k_{iso}$ and further validation is required to avoid biased predictions outside of the ranges considered (Figure 3). A
significant challenge to using this relationship to predict $k_{iso}$ is likely the availability of suitable pedotransfer functions, particularly for $NO_3^-$ availability and microbial biomass, to estimate patterns in the proposed drivers (Van Looy et al., 2017). Given the interaction between soil pH and $NO_3^-$ availability (Figure 3 a & b), the absence of such data may not seriously compromise predictions for fertilised agricultural soils which are typically not strongly acidic. However, accurately predicting natural spatial and seasonal variability and the influence of future changes in atmospheric $NO_3^-$ deposition
(DeForest et al., 2004) may be more problematic. For this reason we considered whether more readily available parameters such as soil texture, carbon content and nitrogen content might provide an alternative basis for empirical predictions of $k_{iso}$ (Van Looy et al., 2017). Relationships between these variables and $k_{iso}$ were relatively weak and could only explain a marginal amount of the observed variability. Considering these properties in combination with soil pH yielded clay content as a secondary significant term potentially reflecting a relatively strong co-correlation (Spearman's $\rho = 0.5$) with $NO_3^-$
availability. Soil pH and clay content may provide an alternative empirical approach to predicting $k_{iso}$ when the availability of soil property data is limited.



**Data availability**

The underlying research data is part of European Research Council grant no. 338264 and will be made publicly accessible as part of a combined data product for this grant. The data may be requested from the corresponding author by email.

**Competing interests**

The authors declare that they have no conflict of interest.

**Author contributions**

Conceptualisation - SJ, AK, JO, SW, AC, LC & LW; Formal analysis – SJ & AK; Funding acquisition – JO & LW; Investigation – SJ, AK, SW & AC; Methodology – SJ, AK, JO & SW; Resources: JO, LC & LW; Writing (original draft) –
SJ; Writing (review & editing) – JO, AC & LC.

**Acknowledgements**

This work was funded by the European Research Council (ERC) under the European Union's Seventh Framework Programme (FP7/2007-2013) grant agreement No. 338264, and the Agence Nationale de la Recherche (ANR) grant number
ANR-13-BS06-0005-01. Many thanks to Jorge Curiel-Yuste, Alexandria Correia, Jean-Marc Ourcival, Jukka Pumpanen, Huizhong Zhang, Carmen Emmel, Nina Buchmann, Sabina Keller, Irene Lehner, Anders Lindroth, Andreas Ibrom, Jens Schaarup Sorensen, Dan Yakir, Fulin Yang, Michal Heliasz, Susanne Burri, Penelope Serrano Ortiz, Maria Rosario Moya Jimenez, Jose Luis Vicente, Holger Tulp, Per Marklund, John Marshall, Nils Henriksson, Raquel Lobo de Vale, Lukas Siebicke, Bernard Longdoz, Pascal Courtois, and Katja Klumpp for providing soil from Eurasian sites, Joana Sauze, Ana
Gutierrez, and Bastien Frejaville for facilitating analyses made in France, and Jon Lloyd, Paul Nelson, Niels Munksgaard, Jen Whan, Michael Bird, Chris Wurster, and Hilary Stuart-Williams for facilitating analyses made in Australia.

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





**Tables**


Table 1: Spearman's rank correlation coefficients ($\rho$) for relationships between site mean oxygen isotope exchange rate ($k_{iso}$), soil pH, microbial biomass (MB), $NO_3^-$ availability and $NH_4^+$ availability measured in untreated soils (n = 44). * indicates p < 0.05 and ** indicates p < 0.01.

|  | $k_{iso}$ | pH | MB | $NO_3^-$ | $NH_4^+$ |
|---|---|---|---|---|---|
| $k_{iso}$ | - | 0.58** | 0.16 | −0.25 | −0.62** |
| pH | 0.58** | - | −0.27 | 0.01 | −0.73** |
| MB | 0.16 | −0.27 | - | 0.29 | 0.05 |
| $NO_3-$ | −0.25 | 0.01 | 0.29 | - | 0.11 |
| $NH_4^+$ | −0.62** | −0.73** | 0.05 | 0.11 | - |

Table 2: Spearman's rank correlation coefficients ($\rho$) for relationships between changes in the ratio of mean rate of oxygen isotope exchange ($k_{iso}$), soil pH, microbial biomass (MB), $NO_3^-$ availability and NH4+ availability between soils receiving a $NH_4NO_3$ addition and that of the corresponding untreated soils (n = 14). * indicates p < 0.05 and ** indicates p < 0.01.

|  | $k_{iso}$ | pH | MB | $NO_3^-$ | $NH_4^+$ |
|---|---|---|---|---|---|
| $k_{iso}$ | - | 0.57* | 0.37 | −0.84** | 0.14 |
| pH | 0.57* | - | 0.22 | −0.75** | 0.02 |
| MB | 0.37 | 0.22 | - | −0.32 | 0.18 |
| $NO_3-$ | −0.84** | −0.75** | −0.32 | - | 0.09 |
| $NH_4^+$ | 0.14 | 0.02 | 0.18 | 0.09 | - |




**Figures**

Figure 1: Theoretical calculations of the expected relationship between the rate of hydration ($k_h$) and hydroxylation reactions, the speciation of dissolved organic carbon (DIC) and the rate of oxygen isotope exchange ($k_{iso}$): a) expected variations in the rate of hydration ($k_h$) and hydroxylation reactions with pH at 21 °C calculated following Uchikawa & Zeebe (2012) and Sauze et al. (2018). Dashed lines indicate uncatalysed rates whilst solid lines include the presence of 200 nM of carbonic anhyrdrase with a $k_{cat}/k_m = 3 \times 10^7$ M s$^{-1}$ and a pka of 7.1. The catalysed rate of hydration decreases under acidic conditions as high proton concentrations limit enzyme regeneration, b) Speciation of dissolved organic carbon (DIC) calculated from rate constants at 21 °C, c) Expected variations in the rate of isotope exchange ($k_{iso}$) with pH calculated as in the first panel (a). The rate of exchange is limited by enzyme regeneration under acidic conditions and the availability of $CO_2$ under alkaline conditions.


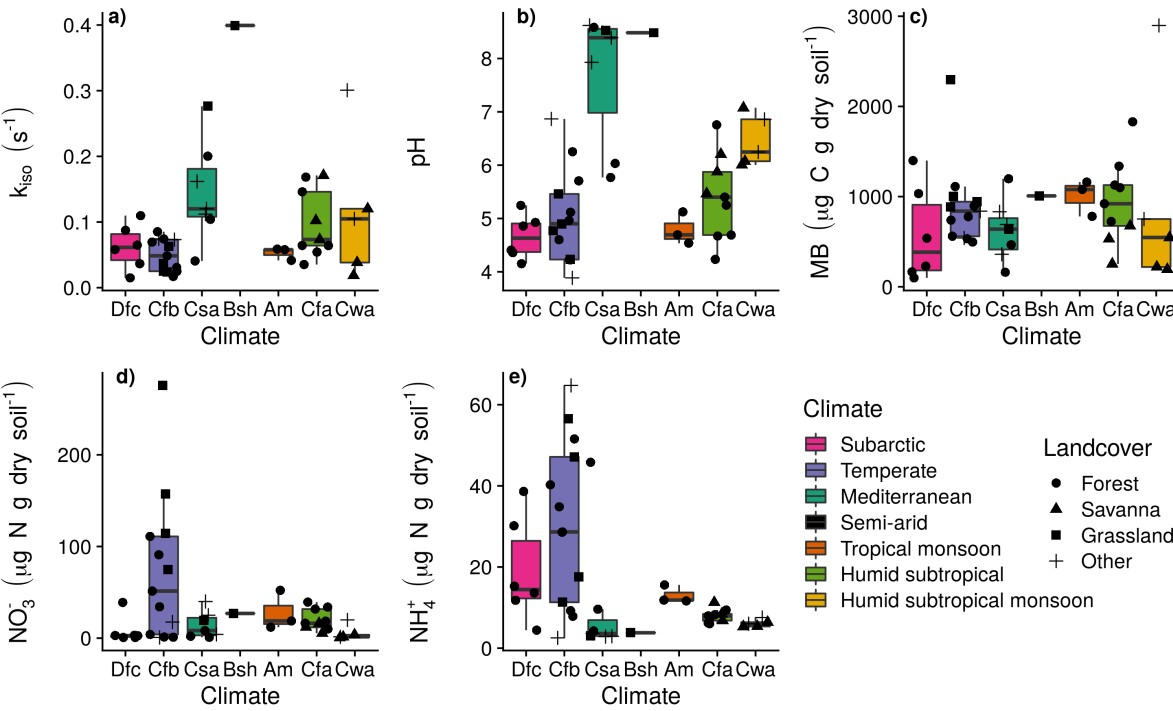

Figure 2: Measurement summaries of mean untreated soils by Köppen-Geiger climatic zone of the sampling site. The 27 sites in western Eurasian (EUR) were within Subartctic (Dfc; n= 6 ), Temperate oceanic (Cfb; n= 13), Hot-summer Mediterraean (Csa; n = 7) and Hot semi-arid (Bsh; n = 1) climate zones and the 17 sites in north Queensland, Australia (AUS) were within Tropical monsoon (Am; n = 3), Humid subtropical (Cfa; n = 9) and Monsoon-influenced humid subtropical (Cwa; n= 5) climate zones. Box lower, middle and upper hinges respectively indicate 0.25, 0.5 and 0.75 quantiles. Over-plotted points are the associated site means (n=2 or 3) with shape indicating land-cover: a) $k_{iso}$, b) pH, c) microbial biomass (MB), d) $NO_3^-$ availability, and d) $NH_4^+$ availability.



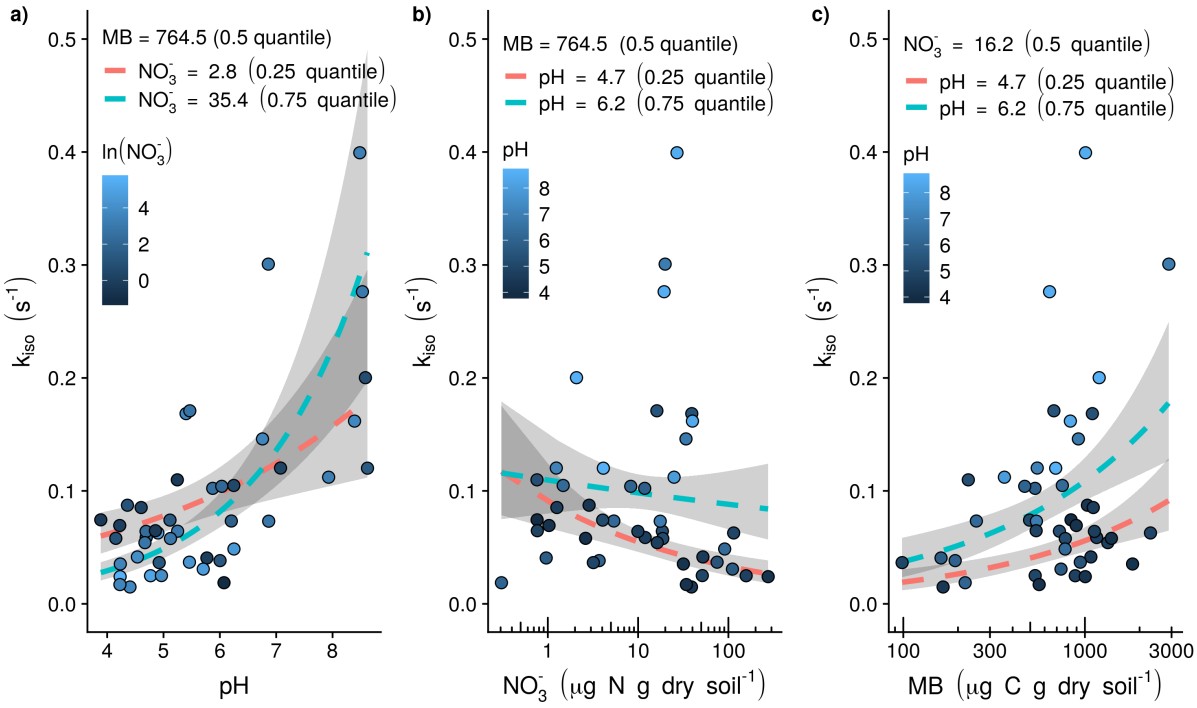

Figure 3: Observed (points) and modelled relationships following Eq. 6 (dashed lines) between the rate of oxygen isotope exchange ($k_{iso}$) and soil pH, $NO_3^-$ availability and microbial biomass (MB): a) the positive relationship between $k_{iso}$ and soil pH with model response as a function of the shown range in soil pH calculated with median microbial biomass and lower quartile (red dashed line) and upper quartile (blue dashed line) $NO_3^-$ availability, b) the negative relationship between $k_{iso}$ and $NO_3^-$ availability with model response as a function of the shown range in $NO_3^-$ availability calculated with median microbial biomass and lower quartile (red dashed line) and upper quartile (blue dashed line) soil pH, and c) the positive relationship between $k_{iso}$ and microbial biomass with the model response as a function of the shown range in microbial biomass calculated with median soil pH and $NO_3^-$ availability (red dashed line). Grey shaded areas indicate the 95 % confidence intervals associated with model fits.





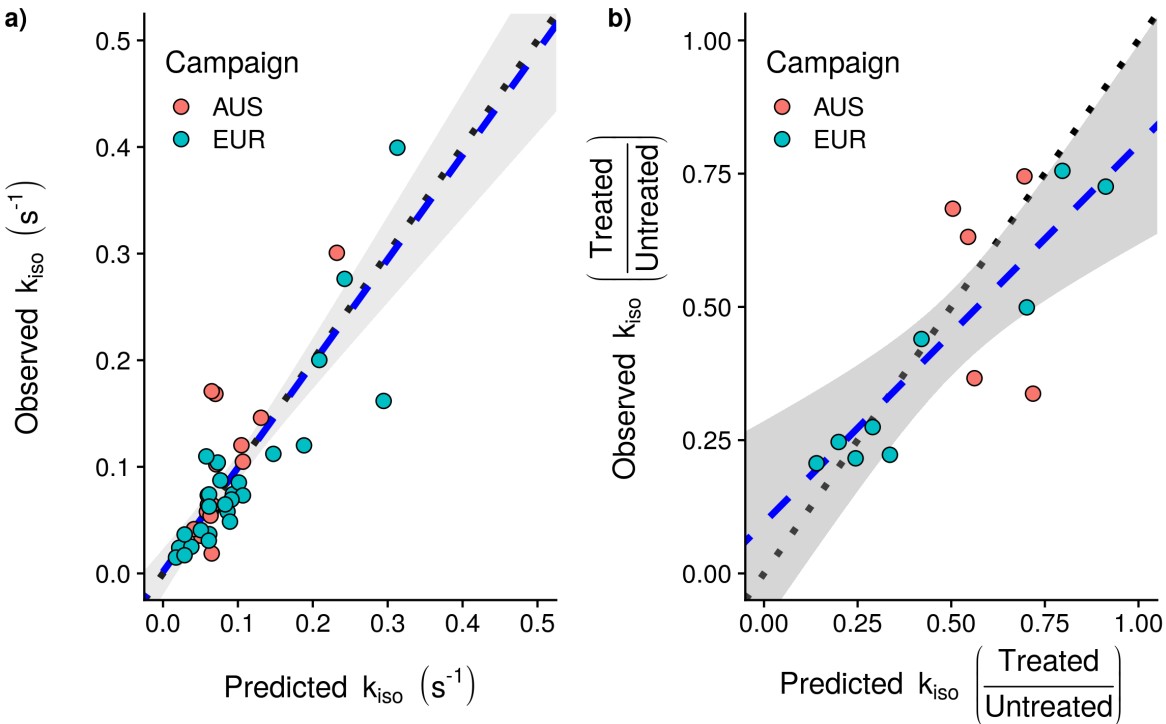

Figure 4: Rates of oxygen isotope exchange ($k_{iso}$) predicted by the minimal adequate model statistical model identified (Eq. 6): a) model predictions for the 44 untreated soils against the observations for these soils used in model fitting and b) predicted fractional changes in $k_{iso}$ between treated and untreated soils against the changes observed following $NH_4NO_3$ addition. Plotted points indicate individual sites with the associate sampling campaign indicated by colour (EUR: blue, n = 9; AUS: red, n = 5), dotted lines indicate the 1:1 line and dashed blue lines indicate linear relationships between predicted and observed values with a shaded 95 % confidence interval.




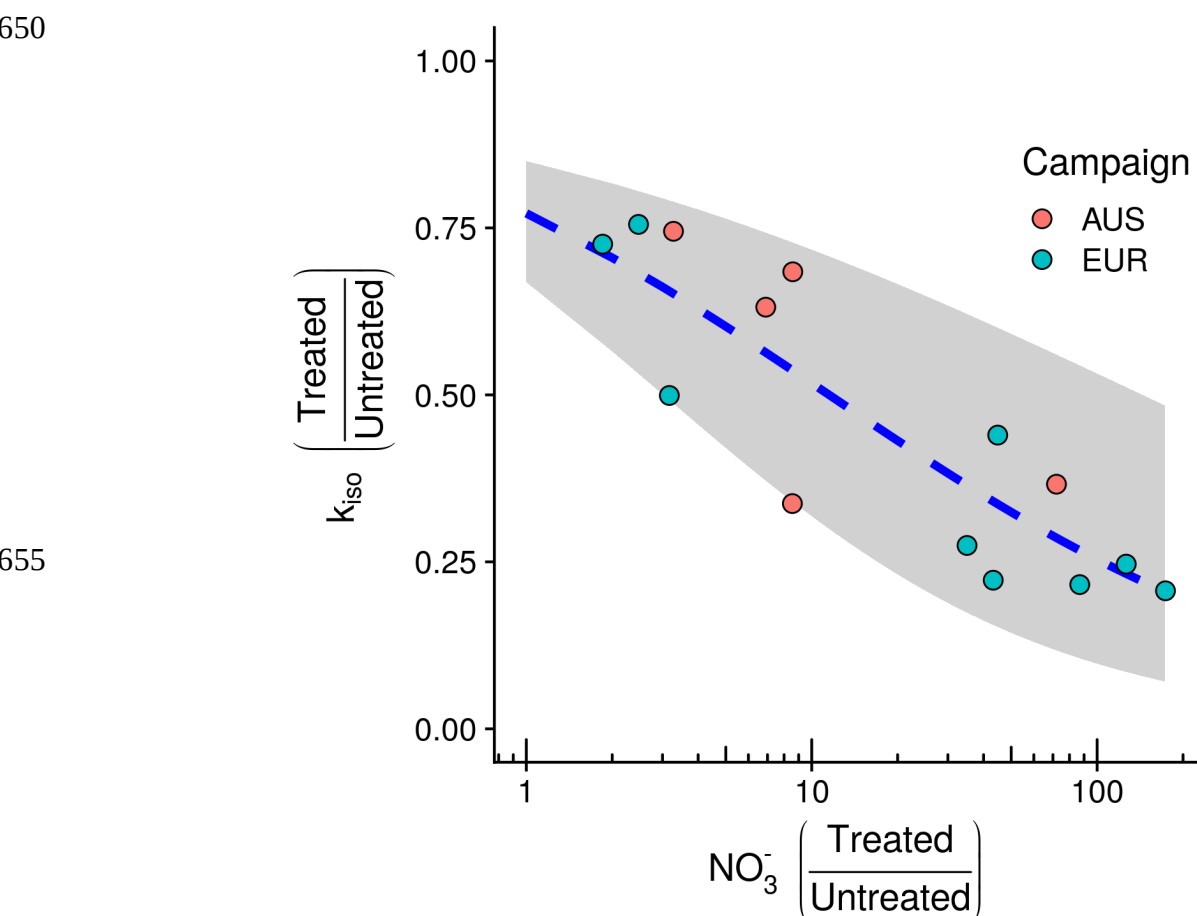

*Figure 5: Observed negative relationship between the fractional change in the rate of oxygen isotope exchange ($k_{iso}$) and the fractional change in $NO_3^-$ availability between treated and untreated soils following $NH_4NO_3$ addition for the 14 sites considered. Plotted points indicate the change for individual sites with the associate sampling campaign indicated by colour (EUR: blue, n= 9; AUS: red, n = 5). On the y-axis, quotients below 1 indicate $k_{iso}$ in soils receiving the treatment decreased relative to corresponding untreated soils for each site. On the x-axis, quotients above 1 indicate $NO_3^-$ availability in soils receiving the treatment increased relative to corresponding untreated soils for each site. The blue dashed line shows the fit of the minimal adequate generalised linear model describing the change in $k_{iso}$ with 95 % confidence intervals shaded in grey.*