# Peer review of "Oxygen isotope exchange between water and carbon dioxide in soils is controlled by pH, nitrate and microbial biomass through links to carbonic anhydrase activity"

_SOIL, 2020_

## Referee Comment (RC1) · Anonymous Referee #1 · 19 Oct 2020

**General comments**

This study reports the rates of oxygen isotope exchange between soil pore-space water and the ambient CO2, $k_\text{iso}$, from a large set of soil samples ($n = 44$) collected from seven ecoclimatic zones in Eurasia and Australia. $k_\text{iso}$ is an essential parameter for quantifying soil influences on the $\delta^{18}\text{O}$ signature of atmospheric CO2. The variability of $k_\text{iso}$ with soil and ecoclimatic features is poorly understood due to the paucity of data and the lack of standardization in measurement and reporting. This study marks a

valuable contribution as it shows that soil pH and $NO_3^-$ content are the most important factors controlling $k_{iso}$ variability. The study may be considered for publication after concerns are addressed.

My main concern is with the interpretation of the results of ammonium nitrate ($NH_4NO_3$) treatment. The authors attributed the decrease of $k_{iso}$ following $NH_4NO_3$ addition to the inhibition of carbonic anhydrase caused by $NO_3^-$. However, other possible mechanisms, namely, inhibition through increased ammonium content or decreased pH cannot be ruled out by the experimental design, nor by the statistical analysis that follows. In essence, $NH_4NO_3$ addition may affect $k_{iso}$ through these causal pathways:

- $NH_4NO_3$ addition $\rightarrow$ $[NH_4^+]$ increase $\rightarrow$ $k_{iso}$ decrease

- $NH_4NO_3$ addition $\rightarrow$ $[NH_4^+]$ increase $\rightarrow$ pH decrease $\rightarrow$ $k_{iso}$ decrease

- $NH_4NO_3$ addition $\rightarrow$ $[NO_3^-]$ increase $\rightarrow$ $k_{iso}$ decrease

To accept Hypothesis 3, the authors must show evidence that *after controlling for all confounding variables*, including pH and $[NH4^+]$, there is still a robust decrease of $k_{iso}$ with the increase of $[NO_3^-]$. Given the absence of a randomized design and the small sample size ($n = 14$) for $NH_4NO_3$ addition treatment, it is difficult to identify $[NO_3^-]$ as the unique cause for carbonic anhydrase inhibition. One possible solution could be to treat pH and $[NH_4^+]$ as instrumental variables, but this would require them to show strong correlation with $[NO_3^-]$. The best way would be to separate different causes through experimental design.

A minor concern I have is that this study was partially motivated by the use of $\delta^{18}O$ of CO2 to estimate terrestrial photosynthesis. While the validity of this method has been demonstrated at the global scale by Welp et al. (2011), I would caution that it is unclear whether the current in-situ observational network would provide sufficient data to resolve regional-scale photosynthesis. Nevertheless, in my opinion, soil–atmosphere

CO2 isotope exchange is an interesting topic for its own sake, regardless of whether $\delta^{18}O_{CO_2}$ can provide constraints on terrestrial photosynthesis with accuracy and spatiotemporal resolution as high as those of other photosynthetic tracers in vogue (e.g., solar-induced chlorophyll fluorescence).

The writing needs more clarity and conciseness. As a rule of thumb, try not to make sentences more complicated than the ideas they convey. In a paragraph, stick to one point and avoid switching topics or walking back and forth. For example, much of the discussion had the main points hidden in the middle of a paragraph and could use some restructuring. Break long paragraphs if necessary.

The hypotheses need to be accurately framed. Hypothesis 2 is a complicated statement, and the only part *testable* based on your experiments is that $k_{iso}$ increases with soil pH. The rest of Hypothesis 2 describes possible mechanisms and they cannot be answered by your experiments. In Hypothesis 3, you can only test whether $k_{iso}$ increases with $[NO_3^-]$, but not whether $[NO_3^-]$ binds carbonic anhydrases or *how* it inhibits carbonic anhydrase. These two hypotheses should be precisely worded as testable hypotheses. The hypotheses you actually tested were stated in P14L382–383, so why not simplify them just like that?

Finally, I encourage the authors to make the data sets publicly available in a data repository. This would make the study more easily discoverable and facilitate data reuse in future studies, for example, comparison across sites and parameterization of related soil processes in a land biosphere model.

**Specific comments**

- P1L13: "The expression and activity of carbonic anhydrase [. . . ]" - You may need to tell the reader that carbonic anhydrase regulates the hydration of CO2 in soil pore-space water before you mention that it drives $k_{iso}$.

- P1L19–20: "[. . . ] potentially reflecting the direct or indirect inhibition of carbonic anhydrases" - Is there a way to tell which mechanism is more likely?

- P2L31: "because the $\delta^{18}O$ of leaf–atmosphere CO2 exchange tends to be enriched [. . . ]" - More precisely, this is because leaf preferentially uses lighter isotopologues of CO2, which diffuse faster than heavier ones. See Farquhar et al. (1993) *Nature* (https://doi.org/10.1038/363439a0).

- P2L44: "Comprising at least six distinct families, [. . . ]" - There are seven now, with the newly discovered $\iota$-CA in phytoplanktons. See Jensen et al. (2019) *ISME J* (https://doi.org/10.1038/s41396-019-0426-8).

- P3L81–82: "Whilst the sensitivity of soil $k_{iso}$ to the presence of specific functional groups, like phototrophs which employ carbonic anhydrases in their carbon concentration mechanisms [. . . ]" - Are phototrophs abundant in soil microbial communities?

- P4L99: Be specific about "the inorganic nitrogen chemistry of soil solutions."

- P5L133–134: Does sieving affect carbonic anhydrase activity in soils?

- P7L195–198: What was the precision of the IRIS for CO2 and $\delta^{18}O_{CO_2}$ measurements when averaged in 40 intervals?

- P7L210: Eq. (1) requires a steady-state condition. What is the turnover time for gas exchange in the cuvette? Could you show that the measurement period (12, P1L191) is much longer than this turnover time?

- P8L238–239: Please considering providing a table of site information and soil characteristics, either as a supplementary table or a metadata file in the online data set associated with this study. Although such information is available for European sites in Kaisermann et al. (2018) *ACP*, it would not be convenient for

the reader to reference across multiple publications. For the Australian sites, I do not see any such data.

- P10L281: What does the "two-term model" mean? What are the predictors?

- P10L282: Have soil texture, carbon content, and nitrogen content been considered in the aforementioned model selection procedures?

- P11L305: "Correlations between all other variable pairings were weaker and non-significant ($p > 0.05$)." - I find this observation in apparent conflict with the interpretation of $NH_4NO_3$ treatment results. If $NO_3^-$ concentration does not control $k_{iso}$ in natural soils, why would adding $NH_4NO_3$ cause $k_{iso}$ to decrease through carbonic anhydrase inhibition? One possible scenario could be that the variation in $k_{iso}$ that is attributable to soil pH is so large that any influence from $NO_3^-$ concentration is obscured. To test whether this would be the case, Spearman's rank correlation would be insufficient. You would need to control for the variation due to pH before testing the effect of $[NO_3^-]$.

- P13L357: While the fraction of explained deviance is high, this is a small sample with $n = 14$ and uncertainty associated with the model could be large. What is the confidence interval of the coefficient of $\ln\left[NO_3^-\right]$?

- P13L376–380: "Whether the potential [. . . ] remains an unresolved but key question." - Not sure what you are trying to mean with this sentence. Please clarify it.

- P15L425: "The absence of strong patterns with climate or land-cover in this study may well reflect the fact that the temperature and moisture conditions used are unrepresentative of field conditions especially for colder and drier sites." - Or, it could also be that soil texture and composition are the main controls.

- P15L435: What are the "pedotransfer functions?"

**Technical comments**

- P1L10: "gross primary *production*" vs. P1L25 "gross primary *productivity*" (emphases mine), pick one.

- P1L11: "ecosystem-scale" → "ecosystem scale"

- P1L15: Add a comma before "indicating [. . . ]."

- P1L33: "the leaves of plants" → "leaves". Pleonasm.

- P2L35: "causing CO2 that interacts with a leaf but is not fixed to inherit the isotopic composition of the leaf water pool" - A difficult sentence. Please clarify.

- P2L44–P3L73: This paragraph has a lot to unpack. In my opinion, to bring clarity to this paragraph, you may consider splitting it into two. Describe the abiotic reaction of oxygen isotope exchange first, and then introduce the role of carbonic anhydrases in accelerating the reaction towards equilibrium. I would consider splitting the paragraph at line 62 and rearraging sentences for a clean separation.

- P3L83: "it's" → "its"

- P3L87–89: "Such an observation may result from changes in size or composition of the microbial communities involved as discussed (Sauze et al., 2017, 2018)." - This is a reiteration of P3L79–81.

- P4L95: "non-carbon" → "non-carbonate"

- P5L123: "principle" → "principal"

- P5L124: "indicted" → "indicated"

- P6L171: This should be section 2.2, not 2.1.

- P11L312–316 and P12L330–337: It is inconvenient to track which model is which. Please consider listing model diagnostics in supplementary tables.

- Figure 3: It is difficult to distingush high values from low values indicated by the color bars. Try to increase the contrast.

- Figure S1: Remove the ocean background and other unnecessary information. Please simplify this figure to make the ecoclimatic classification more evident. Consider putting the legend outside of the figure canvas to avoid interference.

---

## Short Comment (SC1) · 28 Oct 2020

Sam Jones and co-authors,

I have read your work with great interest. The exchange of oxygen isotopes between $CO_2$ and soil water is an important process for $\delta^{18}O$, and this work contributes to a better understanding of that exchange. However, this exchange is also of great importance for the budget of $\Delta^{17}O$ in $CO_2$, a different tracer for GPP.

[Figure]

$\Delta^{17}O$ in $CO_2$ was first proposed as a tracer of GPP by Hoag et al. (2005). More recently, laboratory studies confirmed the effect of photosynthesis on $\Delta^{17}O$ in $CO_2$ (Adnew et al., 2020), and we simulated large-scale variations of $\Delta^{17}O$ in atmospheric $CO_2$ (Koren et al., 2019). We struggled with representing the soil exchange in that model, and for follow-up studies we can possibly improve our representation of soil exchange using Eq. 6 from your manuscript.

I think you can reach a greater audience if you also explicitly address the $\Delta^{17}O$ community in your work.

Some other suggestions/questions:

- In the first line and last line of the abstract I would replace "$\delta^{18}O$" with "$\delta^{18}O$ and $\Delta^{17}O$".

- Sec 2. Are you sure that the sampling and transporting of soil samples does not affect the CA or microbes in the sample?

- There are two sections with number 2.1.

- L139: "Tillburg". This should be the lovely city "Tilburg" .

- L147: Why did you choose to report on the $VPDB_g$ scale, instead of e.g. VSMOW?

- L210: The units provided in the text do not agree with Eq. 1 .

- L423: I would briefly mention $\Delta^{17}O$ here.

- Caption Fig. 1: The authors mention twice: "dissolved organic carbon (DIC)". Should this be DIC or DOC?

References:

Adnew et al. (2020). Leaf-scale quantification of the effect of photosynthetic gas exchange on $\Delta^{17}O$ of atmospheric $CO_2$. *Biogeosciences*. https://doi.org/10.5194/bg-17-3903-2020

Hoag et al. (2005). Triple oxygen isotope composition of tropospheric carbon dioxide as a tracer of terrestrial gross carbon fluxes. *Geophysical Research Letters*. https://doi.org/10.1029/2004GL021011

Koren at al. (2019). Global 3-D Simulations of the Triple Oxygen Isotope Signature $\Delta^{17}O$ in Atmospheric $CO_2$. *Journal of Geophysical Research: Atmospheres*. https://doi.org/10.1029/2019JD030387

---

## Editor Comment (EC1) · Steven Sleutel (Editor) · 15 Jan 2021

Steven Sleutel (Editor)

steven.sleutel@ugent.be

SOIL-2020-44 presents on an interesting research into how soil factors would affect O18 exchange between CO2 and H2O. As the 18O levels of emitted CO2 are used for estimating land-atmosphere carbon exchange it presents a well-delineated and timely topic. The experimental assessment of exchange of 18O between soil water and CO2 is by no means trivial, yet as many as 44 soils collected from a wide geographical area (Eurasia and Australia) were assessed for their potential to oxygen isotope exchange. The description was mostly excellent but I concur with the referee that on instances the

text is lengthy. For instance the description of the statistical approach and reporting of fitted general linear models are longer than usual. However, here such an elaborate formulation appeared warranted given the clear collinearity of some of the predictor variables. Still, in particular the introduction can surely be further condensed. Also the lengthy description of the setup to administer air with CO2 with contrasting $\delta$18O requires shortening (e.g. by just referring to or some part of it should be moved to supplementary material. The description of preparation of soil mesocosms could also be condensed. Please also use a different term for available NO3- and available NH4+ (or 'availability of'). Both were measured in typical 1M KCl extracts and are best referred to exchangeable NO3- and NH4+. The entire text is filled with long complex sentences, which on their own are well crafted but do not always allow fluid reading. I would recommend the authors to read their manuscript once more and if they see fit do try to subdivide some of the longer sentences. I concur with the referee that some of the hypotheses require rephrasing and attention needs to be given to his/her possibly justified concern on the NH4NO3 administration's ineptitude to robustly prove that NO3- led to inhibition of of carbonic anhydrase. There are also two other issues that need to be addressed before publication can be considered:

1° The interpretation of the relationship between NH4+ and kiso: On L408 following statement is made 'Notably the weak relationship between changes in kiso and NH4+ availability identified in this experiment (Table 2) suggests the relationship between these variables across the untreated soils (Table 1) does indeed reflect the pH sensitivity of ammonia speciation rather than a direct causal link.'. I would agree that the link is indeed direct, but strongly doubt that NH4+/NH3 speciation is important here. Below pH 7 there is virtually no (toxic) NH3. Most soils had acidic pH so this explanation seems wrong. Instead: high NH4+ levels in upland soils rather suggest nitrification is impeded in some way. There is an obvious link between pH and NH4+ levels: at low pH, nitrification is well known to be slowed: indeed there was a strong negative correlation between both (Table 1) and so the negative correlation between NH4+ and kiso may just be indirect through their mutual relation with pH. Alternatively, higher NH4+

levels point at impeded autotrophic nitrification – a rather energetically unfavourable process and therefore sensitive to environmental constraints. Inhibited nitrification may also point at unfavourable conditions for other microbial processes: perhaps also production of anhydrase activity? In Table 2 the artificially elevated $NH4+$ levels are no longer the resultant from slow or fast pH-dependent nitrification and therefore do not display any relation with kiso. The referee also commented on this matter– please do take that comment into account.

$2°$ It is unfortunate that the authors opted for an enourmous dose of $NH4NO3$, viz 0.7 mg $NH4NO3$ g-1 (L168). No doubt such a large addition of $NH4+$ would have led to serious soil acidification following its partial nitrification during the 1 week pre-incubation at $23°C$. And on the other hand the obtained $NH4+$ and $NO3-$ levels in treated soil would not reflect environmentally realistic exchangeable N levels. That would severely limit the relevance of Fig. 5. Or is the 0.7 mg $NH4NO3$ g-1 a typing error? In any case in their rebuttal the authors will need to present (not necessarily in the revised manuscript) the absolute increase in $NH4+$ and $NO3-$ levels alongside changes in pH brought forth by the $NH4NO3$ administration as based on the fractorial changes in $NO3-$ now presented it is impossible to judge whether or not excessive amounts of N had been added. This firstly needs to be clarified.

Minor comment: Remove commas when the citation is part of the sentence throughout the entire text: Jones et al. (2017), not Jones et al., (2017).

L21 suggest to omit 'future'

L31 'the $δ18O$ of leaf-atmosphere CO2 exchange' reads strange, sounds clearer without 'exchange'

L 35 CO2 that interacts with a leaf – interacts seems too vague, could this be reworded?

L36 'undergo considerable enrichment' of what?

L40 'alters the $δ18O$ of atmospheric CO2' was not really clear to me. Perhaps write

'alters the $δ18O$ of CO2 in soil vs. atmospheric CO2'?

L51 is 'of soil atmospheric CO2' not better?

L56 'abiotically invades' sounds awkward, perhaps write 'diffuses into the soil pore network'

L65 delete 'true', is confusing here

L99 better 'agricultural soils'?

L166 'an additional three replica incubations' correct English?

2.1 Not clear if soils were kept cool during transport

L273 remove 'presumably reflecting the pH dependency of $NH4+$ and ammonia speciation' this does not belong in the M&M section

L293-294 do not seem entirely accurate: there seems to be a higher kiso for the Csa group.

L370-375 is quite unclear: are the 0.04 to 13 s-1 field values derived from your lab measured 0. To 0.4 s-1 kiso estimates? + which older studies. The whole part starts quite sudden. In fact for the sake of clarity the apparent issue of comparing lab and field estimates is perhaps best omitted from the paper entirely – also further on.

L431 'between untreated and treated' is not a very clear phrasing – spell out better what you are referring at.

L434 'A significant challenge to using this relationship to predict kiso is likely the availability of suitable pedotransfer functions,particularly for $NO3-$ availability and microbial biomass, to estimate patterns in the proposed drivers (Van Looy et al., 2017).' Is an understatement. It will be impossible to predict the ephemeral soil $NO3-$ levels with a simple pedotransfer function. The authors best refer to use soil N models to predict NO3 levels and use these as inputs into Eq 6.

---

## Author Comment (AC1) · 13 Feb 2021

Response to Gerbrand Koren

We would like to thank Gerbrand for taking the time to read and helpfully comment on this manuscript. We have responded to their comments (reproduced in blue) below.

1) I have read your work with great interest. The exchange of oxygen isotopes between $CO_2$ and soil water is an important process for $\delta^{18}O$, and this work contributes to a better understanding of that exchange. However, this exchange is also of great importance for the budget of $\Delta^{17}O$ in $CO_2$, a different tracer for GPP. $\Delta^{17}O$ in $CO_2$ was first proposed as a tracer of GPP by Hoag et al. (2005). More recently, laboratory studies confirmed the effect of photosynthesis on $\Delta^{17}O$ in $CO_2$ (Adnew et al., 2020), and we simulated large-scale variations of $\Delta^{17}O$ in atmospheric $CO_2$ (Koren et al., 2019). We struggled with representing the soil exchange in that model, and for follow-up studies we can possibly improve our representation of soil exchange using Eq. 6 from your manuscript. I think you can reach a greater audience if you also explicitly address the $\Delta^{17}O$ community in your work.

Thank you very much for your positive comment. We appreciate your interest in our work and agree with you that our work could also interest groups like yours working on the $\Delta^{17}O$ in $CO_2$. We have now added a couple of sentences in the introduction to clarify this point.

2) In the first line and last line of the abstract I would replace "$\delta^{18}O$" with "$\delta^{18}O$ and $\Delta^{17}O$".

We have rephrased the first and last line of the abstract to remove the emphasis on only $\delta^{18}O$ and replaced this with a more general reference to the oxygen isotope composition of atmospheric $CO_2$. In addition we have referenced explicitly in the introduction text the importance of $k_{iso}$ for understanding the $\delta^{17}O$ composition of $CO_2$ and the $\Delta^{17}O$ of $CO_2$.

3) Sec 2. Are you sure that the sampling and transporting of soil samples does not affect the CA or microbes in the sample?

We do indeed expect there to be a disturbance effect on the microbial community when transporting soils and sieving them, thus it is important to be mindful of this when comparing results from soils measured under field conditions and those measured in laboratory experiments as well as extrapolating results from mesocosms to the large scale. This study however was designed to characterize a set of homogenized climate-controlled soils to make a link between the measured CA activity, the mesocosm soil characteristics and their response to changes in inorganic N concentrations. However the quantitative influence of transport and sieving on carbonic anhydrase activity is so far not well understood but is discussed. Please see L360 - L380 in the Discussion.

4) There are two sections with number 2.1.

Thanks. We have corrected section '2.1 Gas exchange measurements' to be '2.2 Gas exchange measurements'.

5) L139: "Tillburg". This should be the lovely city "Tilburg".

Thanks. We have corrected this.

6) L147: Why did you choose to report on the VPDBg scale, instead of e.g. VSMOW?

We preferentially report our $CO_2$ in air measurements on the VPDBg scale (also known as VPDB-$CO_2$ scale) reflecting the fact that values assigned to our working standards are ultimately tied to the acid digestion of RM NBS-19 calcite. Please see:

Werner, R. A., Rothe, M. and Brand, W. A.: Extraction of $CO_2$ from air samples for isotopic analysis and limits to ultra high precision $\delta^{18}O$ determination in $CO_2$ gas, Rapid Communications in Mass Spectrometry, 15(22), 2152–2167, doi:https://doi.org/10.1002/rcm.487, 2001.

Werner, R. A. and Brand, W. A.: Referencing strategies and techniques in stable isotope ratio analysis, Rapid Communications in Mass Spectrometry, 15(7), 501–519, doi:https://doi.org/10.1002/rcm.258, 2001.

7) L210: The units provided in the text do not agree with Eq. 1.

Thanks! We have removed the erroneous "m$^{-3}$" from "where u is the flow rate (mol s$^{-1}$) through the chamber line".

8) L423: I would briefly mention Δ17O here.

Please see our response to comment 2) above.

9) Caption Fig. 1: The authors mention twice: "dissolved organic carbon (DIC)". Should this be DIC or DOC?

Thanks, this should indeed be dissolved inorganic carbon! We have altered the caption accordingly.

---

## Author Comment (AC2) · 13 Feb 2021

Response to Anonymous Referee #1

We would like to thank the referee for taking the time to review this manuscript. Their comments helped us to greatly improve the manuscript. You will find a point-by-point response to these comments (reproduced in blue) below.

1) My main concern is with the interpretation of the results of ammonium nitrate (NH4NO3) treatment. The authors attributed the decrease of kiso following NH4NO3 addition to the inhibition of carbonic anhydrase caused by NO3-. However, other possible mechanisms, namely, inhibition through increased ammonium content or decreased pH cannot be ruled out by the experimental design, nor by the statistical analysis that follows.

In essence, NH4NO3 addition may affect kiso through these causal pathways:

- NH4NO3 addition  $\rightarrow$  [NH4+] increase  $\rightarrow$  kiso decrease
- NH4NO3 addition  $\rightarrow$  [NH4+] increase  $\rightarrow$  pH decrease  $\rightarrow$  kiso decrease
- NH4NO3 addition  $\rightarrow$  [NO3-] increase  $\rightarrow$  kiso decrease

To accept Hypothesis 3, the authors must show evidence that after controlling for all confounding variables, including pH and [NH4+], there is still a robust decrease of kiso with the increase of [NO3-]. Given the absence of a randomized design and the small sample size (n=14) for NH4NO3 addition treatment, it is difficult to identify [NO3-] as the unique cause for carbonic anhydrase inhibition. One possible solution could be to treat pH and [NH4+] as instrumental variables, but this would require them to show strong correlation with [NO3-]. The best way would be to separate different causes through experimental design.

We agree that the experimental design of the ammonium nitrate treatment is not sufficient to fully tease apart the combined systematic effects (i.e. increased nitrate and ammonium availability and decreased soil pH) of the treatment on  $k_{iso}$ . As the reviewer states, a more extensive controlled factorial experiment would be required to achieve this. However, the results of this experiment (Section 3.2; Figure 5; Table S3), that show changes in  $k_{iso}$  are most strongly linked to changes in nitrate availability (pathway 3 above) and to a lesser degree soil pH (pathway 2 above) but do not appear related to changes in ammonium availability (pathway 1 above), are still informative to the interpretation of the wider study. Across the untreated soils we clearly identify soil pH, nitrate availability and microbial biomass as explaining variations in  $k_{iso}$  (Section 3.1; Figure 3; Table S1). Agreement between the results of both these analyses helps reinforce the importance of pH and nitrate (pathways 2 and 3) but not, directly at least, a role for ammonium (pathway 1). We have adjusted the text in the abstract and Section 4 to acknowledge that limitations of the experimental treatment prevent the definite conclusion that only nitrate, and not some combination of effects, influences the decrease in  $k_{iso}$  observed following the fertilisation treatment.

Abstract: "This effect appears to be supported by a supplementary ammonium nitrate fertilisation experiment conducted on a subset of the soils"

Section 4: "It is important to note that whilst the relationship between the changes in  $k_{iso}$  and exchangeable  $NO_3^-$  are supported by observations from the untreated dataset, the experimental design used in this addition experiment is not sufficient to fully test the influence of the combined changes in soil pH, exchangeable  $NO_3^-$ , exchangeable  $NH_4^+$  and microbial biomass on  $k_{iso}$ . Further controlled, factorial experiments are needed for this purpose."

2) A minor concern I have is that this study was partially motivated by the use of  $\delta$ 18O of CO2 to estimate terrestrial photosynthesis. While the validity of this method has been demonstrated at the global scale by Welp et al. (2011), I would caution that it is un-clear whether the current in-situ observational network would provide sufficient data to resolve regional-scale photosynthesis. Nevertheless, in my opinion, soil–atmosphere CO2 isotope exchange is an interesting topic for its own sake, regardless of whether  $\delta$ 18O-CO2 can provide constraints on terrestrial photosynthesis with accuracy and spatio-temporal resolution as high as those of other photosynthetic tracers in vogue (e.g., solar-induced chlorophyll fluorescence).

We agree with the reviewer that the current in-situ observational network of  $\delta^{18}$ O in atmospheric CO2 is rather coarse, at least compared to the network for total CO2 mixing ratio. However, there are still more than 50 atmospheric stations measuring  $\delta^{18}$ O in CO2, spread across all latitudes and continents, with some of them covering several decades of measurements. The extremely large north-south gradient of  $\delta^{18}$ O in CO2 and its seasonal and interannual dynamics brings unique information on the seasonality and inter-annual variability of the northern hemisphere CO2 sink, which is the strongest land carbon sink at the global scale and with the largest long-term trend (Ciais et al. 2019). Currently, this information is obscured by the lack of understanding of how soil dwelling organisms (and their carbonic anhydrase activity) affect this signal (Wingate et al. 2009). This study presents the largest soil dataset ever gathered on soil  $\delta^{18}$ O-CO2 exchange. The in-depth analysis of the drivers of soil carbonic anhydrase activity that this study brings also serves as an important stepping-stone to study other emerging tracers of the carbon cycle including the  $\Delta^{17}$ O anomaly in CO2 (Koren et al. 2019) and COS (Campbell et al., 2017). For all these reasons, it would seem awkward not to mention the implication this study will have in the future development of independent tracers to study the global carbon cycle. We

also agree that solar-induced chlorophyll fluorescence (SIF), another independent proxy of photosynthesis, has the advantage of being detected from space, conferring a global coverage. However the relationship between SIF detected from space and land photosynthesis is still not well understood, notably in disentangling structural and physiological factors. For this reason, SIF is most interesting at very high spatial resolution, which is only possible since the late 2010s, with satellite instruments like TROPOMI launched in 2017 or FLEX that will be launched in 2022. Here we do not pretend that  $\delta^{18}O$ -CO2 is a more powerful tracer compared to other tracers of global photosynthesis (i.e. SIF or COS), but we are convinced that  $\delta^{18}O$ -CO2 contains unique independent and historical information, strongly linked to the global water and carbon cycle, that cannot be discarded. This study, with its extensive survey of soil types and biomes, addresses one of the key knowledge gaps that currently prevent the routine use of  $\delta^{18}O$ -CO2 as a global carbon tracer, and should motivate the community to reconsider this independent tracer in global climate models, thus constraining our understanding of variability in the northern hemisphere land carbon sink. Hopefully, this may also stimulate the development of a denser observational network of  $\delta^{18}O$  in CO2, which is now possible with the next generation of laser-based CO2 isotope analysers.

Campbell, J. E., Berry, J. A., Seibt, U., Smith, S. J., Montzka, S. A., Launois, T., Belviso, S., Bopp, L. and Laine, M.: Large historical growth in global terrestrial gross primary production, Nature, 544(7648), 84, https://doi.org/10.1038/nature22030, 2017.

Ciais P, Tan J, Wang X et al. (2019) Five decades of northern land carbon uptake revealed by the inter-hemispheric CO2 gradient. Nature, 568, 221–225.

Koren G, Schneider L, van der Velde IR et al. (2019) Global 3-D Simulations of the Triple Oxygen Isotope Signature  $\Delta$ 17O in Atmospheric CO2. Journal of Geophysical Research-Atmospheres, 127, 73.

Wingate L, Ogee J, Cuntz M et al. (2009) The impact of soil microorganisms on the global budget of  $\delta$ 18O in atmospheric CO2. Proceedings of the National Academy of Sciences of the United States of America, 106, 22411–22415.

3) The writing needs more clarity and conciseness. As a rule of thumb, try not to make sentences more complicated than the ideas they convey. In a paragraph, stick to one point and avoid switching topics or walking back and forth. For example, much of the discussion had the main points hidden in the middle of a paragraph and could use some restructuring. Break long paragraphs if necessary.

**Thanks, we have worked to improve the clarity and conciseness of the revised manuscript following this good advice.**

4) The hypotheses need to be accurately framed. Hypothesis 2 is a complicated statement, and the only part testable based on your experiments is that kiso increases with soil pH. The rest of Hypothesis 2 describes possible mechanisms and they cannot be answered by your experiments. In Hypothesis 3, you can only test whether kiso increases with [NO3-], but not whether [NO3-] binds carbonic anhydrases or how it inhibits carbonic anhydrase. These two hypotheses should be precisely worded as testable hypotheses. The hypotheses you actually tested were stated in P14L382–383, so why not simplify them just like that?

We thank the reviewer for this suggestion and we have simplified our hypotheses to reflect the reviewer's comments. This section now reads: "Based on the potential controls on  $k_{iso}$  presented above we tested three specific, non-exclusive, hypotheses; 1)  $k_{iso}$  increases as microbial biomass increases (H1), 2)  $k_{iso}$  increases as soil pH increases (H2), and 3)  $k_{iso}$  decreases as NO3- availability increases (H3)."

5) Finally, I encourage the authors to make the data sets publicly available in a data repository. This would make the study more easily discoverable and facilitate data reuse in future studies, for example, comparison across sites and parameterization of related soil processes in a land biosphere model.

We agree with the reviewers comment and have submitted the dataset (Nov 2020) from this paper to PANGAEA (https://pangaea.de/) for archiving (see also comment 15).

**Specific comments**

6) P1L13: "The expression and activity of carbonic anhydrase [...]" - You may need to tell the reader that carbonic anhydrase regulates the hydration of CO2 in soil pore-space water before you mention that it drives kiso.

Thanks. We have rephrased this sentence as suggested: "As the enzyme carbonic anhydrase enhances the rate of  $CO_2$  hydration within the water-filled pore spaces of soils it is important to develop understanding of how environmental drivers can impact carbonic anhydrase expression and activity and alter  $k_{iso}$ ."

7) P1L19–20: "[...] potentially reflecting the direct or indirect inhibition of carbonic anhydrases" - Is there a way to tell which mechanism is more likely?

To distinguish whether the impact of nitrate is direct or indirect an integrated study looking into changes in the concentration of carbonic anhydrase protein and the abundance of carbonic anhydrase transcripts alongside measurements of  $k_{iso}$  would be required. Additionally it would also be important to do some detailed protein studies that show the physical interaction of nitrate with the carbonic anhydrase protein and develop a method that could quantify the binding efficiency of nitrate to carbonic anhydrase for a few of the dominant soil carbonic anhydrases e.g. the beta-CA class. Collectively these different experiments would help us tease apart the direct and indirect effects of nitrate on carbonic anhydrase in soils.

8) P2L31: "because the  $\delta$ 18O of leaf–atmosphere CO2 exchange tends to be enriched [...]" - More precisely, this is because leaf preferentially uses lighter isotopologues of CO2, which diffuse faster than heavier ones. See Farquhar et al. (1993) Nature (https://doi.org/10.1038/363439a0).

Actually, diffusion is not the only reason. We agree that the oxygen isotope composition of leaf-atmosphere  $CO_2$  exchange is partly explained by fractionation during diffusion, but not only by this. The isotopic exchange between  $CO_2$  and water is also very important (Farquhar et al. 1993). In contrast the influence of oxygen isotope fractionation during other steps of fixation (e.g. carboxylation) is limited because carbonic anhydrase concentrations are sufficiently high enough for the isotopic equilibration between  $CO_2$  and water to be extremely rapid (Ogée et al. 2018). By analogy to 13C fractionation during photosynthesis, Farquhar et al. (1993) described the leaf as consuming isotopically lighter  $CO_2$  in terms of 18O, thereby leaving behind  $CO_2$  enriched in 18O in the intercellular air space to diffuse back to the atmosphere. However, the analogy works because the  $CO_2$  inside the leaf equilibrates its oxygen isotopes with evaporatively enriched leaf water. Thus, the mechanism is very different than for 13C, and primarily driven by leaf water isotopic composition and secondarily by diffusion.

Farquhar, G. D., Lloyd, J., Taylor, J. A., Flanagan, L. B., Syvertsen, J. P., Hubick, K. T., Wong, S. C. and Ehleringer, J. R. (1993) Vegetation effects on the isotope composition of oxygen in atmospheric CO2, Nature, 363(6428), 439–443, doi:10.1038/363439a0.

Ogée J, Wingate L, Genty B (2018) Estimating mesophyll conductance from measurements of C18OO photosynthetic discrimination and carbonic anhydrase activity. Plant Physiol., 178, 728–752.

9) P2L44: "Comprising at least six distinct families, [...]" - There are seven now, with the newly discovered I-CA in phytoplanktons. See Jensen et al. (2019) ISME J (https://doi.org/10.1038/s41396-019-0426-8).

Thanks! We have corrected this and updated the references and manuscript text accordingly. "Comprising at least seven distinct families, carbonic anhydrases have independently evolved in all domains of life in order to catalyse the reversible hydration of carbon dioxide (CO2) to bicarbonate (Jensen et al., 2019)"

10) P3L81–82: "Whilst the sensitivity of soil kiso to the presence of specific functional groups, like phototrophs which employ carbonic anhydrases in their carbon concentration mechanisms [. . . ]" - Are phototrophs abundant in soil microbial communities?

In a review of the literature Wingate et al., 2009 estimated that soil algal populations of between  $10^3 - 10^6$  per gram of soil are typically present in most soils. If cyanobacteria are further included, phototrophs can indeed form an important part of the soil microbial community under many conditions (Muriel Bristol Roach, 1927; Seppey et al., 2017). This may be either as superficial crusts or within the near surface. Whilst they are likely to be less ubiquitous than fungi and bacteria, the possibility of specialised, carbonic anhydrase dependent, carbon concentration mechanisms might suggest their presence could have a disproportionately strong influence on  $k_{iso}$ . In a previous study looking at the role of phototrophs on carbonic anhydrase activity (Sauze et al., 2017) we developed a qPCR approach that helped us show that the putative natural abundance of soil phototrophs derived from the number of 23S reads were relatively small under darkened conditions compared to the bacterial (16S) and fungal (18S) abundances but their relative abundances increased significantly when incubated in the light. This probably and unsurprisingly suggests that such an influence might be somewhat dependent on the canopy cover and light conditions of the system in question.

Muriel Bristol Roach, B. (1927). On the algae of some normal English soils. *The Journal of Agricultural Science*, *17*(4), 563-588. doi:10.1017/S0021859600018839

Seppey, C. V. W., Singer, D., Dumack, K., Fournier, B., Belbahri, L., Mitchell, E. A. D. and Lara, E.: Distribution patterns of soil microbial eukaryotes suggests widespread algivory by phagotrophic protists as an alternative pathway for nutrient cycling, Soil Biology and Biochemistry, 112, 68–76, doi:10.1016/j.soilbio.2017.05.002, 2017.

Sauze, J., Ogée, J., Maron, P.-A., Crouzet, O., Nowak, V., Wohl, S., Kaisermann, A., Jones, S. P. and Wingate, L.: The interaction of soil phototrophs and fungi with pH and their impact on soil CO2, CO18O and OCS exchange, Soil Biology and Biochemistry, 115(Supplement C), 371–382, doi:10.1016/j.soilbio.2017.09.009, 2017.

Wingate L., Ogée J., Cuntz M., B. Genty, I. Reiter, U. Seibt, D. Yakir, K. Maseyk , E.G. Pendall, M.M. Barbour, B. Mortazavi, R. Burlett, P. Peylin, J. Miller, M. Mencuccini, J.H. Shim, J. Hunt, J. Grace (2009) The impact of soil microorganisms on the global budget of  $\delta^{18}$ O in atmospheric CO2. *Proceedings of the National Academy of Sciences of America*, 106, 22411–22415.

11) P4L99: Be specific about "the inorganic nitrogen chemistry of soil solutions."

We have changed this sentence to the following:

"In this respect, the fact that nitrate (NO3-) has also been shown to inhibit carbonic anhydrases (Peltier et al., 1995) suggests that the application of common fertilisers such as ammonium nitrate may exert a considerable control on carbonic anhydrase activity. Indeed, this hypothesis is supported by recent ammonium nitrate fertilising experiments that demonstrated decreases in carbonyl sulphide exchange (Kaisermann et al., 2018b), also catalyzed by carbonic anhydrases, but the influence on  $k_{iso}$  has yet to be considered."

12) P5L133–134: Does sieving affect carbonic anhydrase activity in soils?

Our experiments did not test for the impact of sieving on soil carbonic anhydrase activity and as far as we are aware this has not been reported in the literature, thus the nature of these effects is not well understood and is discussed in L360 to L380.

13) P7L195–198: What was the precision of the IRIS for CO2 and  $\delta$ 18O-CO2 measurements when averaged in 40 intervals?

We have added this information from Jones et al., 2017: "The associated precision for the total concentration and  $\delta^{18}$ O of CO2 was 0.02 ppm and 0.06 % VPDBg respectively."

Jones, S. P., Ogée, J., Sauze, J., Wohl, S., Saavedra, N., Fernández-Prado, N., Maire, J., Launois, T., Bosc, A. and Wingate, L. (2017) Non-destructive estimates of soil carbonic anhydrase activity and associated soil water oxygen isotope composition, Hydrology and Earth System Sciences, 21(12), 6363–6377, doi:https://doi.org/10.5194/hess-21-6363-2017.

14) P7L210: Eq. (1) requires a steady-state condition. What is the turnover time for gas exchange in the cuvette? Could you show that the measurement period (12, P1L191) is much longer than this turnover time?

As in Jones et al. (2017) the turnover time was less than 10 minutes. We have added this information to the text: "The turnover time of air in the jar was less than 10 minutes". Each jar was flushed for 20 or 22 minutes before the measurement period (L189-193) and 22 or 24 minutes before the first used chamber measurement was made. These timings reflect the need to balance the trade-off between approximate steady-state conditions and changes in the isotopic composition of the soil water pool (Jones et al. 2017).

Jones, S. P., Ogée, J., Sauze, J., Wohl, S., Saavedra, N., Fernández-Prado, N., Maire, J., Launois, T., Bosc, A. and Wingate, L. (2017) Non-destructive estimates of soil carbonic anhydrase activity and associated soil water oxygen isotope composition, Hydrology and Earth System Sciences, 21(12), 6363–6377, doi:https://doi.org/10.5194/hess-21-6363-2017.

15) P8L238–239: Please considering providing a table of site information and soil characteristics, either as a supplementary table or a metadata file in the online data set associated with this study. Although such information is available for European sites in Kaisermann et al. (2018) ACP, it would not be convenient for C4 the reader to reference across multiple publications. For the Australian sites, I do not see any such data.

We have submitted the dataset used for archiving in PANGAEA (https://pangaea.de/) and will include the relevant information in the finalised manuscript or as an amendment once the archiving process is complete (see also comment 5).

**16) P10L281: What does the "two-term model" mean? What are the predictors?**

Two-term models are those limited to 2 or less predictive terms. We have rephrased this to make it clearer: "The same approach was also applied to the 27 soils from the EUR sampling campaign and extended to consider the relationships with soil texture and carbon and nitrogen contents to investigate their utility in upscaling efforts. To prevent over-fitting, these models were limited to a maximum of two of predictive terms. The predictive terms considered were soil sand, silt, clay, carbon and nitrogen content, the ratio of carbon to nitrogen content and soil pH."

17) P10L282: Have soil texture, carbon content, and nitrogen content been considered in the aforementioned model selection procedures?

Yes, the same model selection procedures were used. Please see previous comment where this is now explicitly stated in the text.

18) P11L305: "Correlations between all other variable pairings were weaker and non-significant (p > 0.05)." - I find this observation in apparent conflict with the interpretation of NH4NO3 treatment results. If NO3- concentration does not control kiso in natural soils, why would adding NH4NO3 cause kiso to decrease through carbonic anhydrase inhibition? One possible scenario could be that the variation in kiso that is attributable to soil pH is so large that any influence from NO3- concentration is obscured. To test whether this would be the case, Spearman's rank correlation would be insufficient. You would need to control for the variation due to pH before testing the effect of [NO3-].

Spearman's rank correlation is used to identify the strongest patterns between pairs of variables without making a priori assumptions about the data. This is particularly useful as it helps us identify potential co-correlations such as that between pH and ammonium availability that may confound the subsequent analyses discussed in the paragraph following that referred to in this comment.

Subsequent use of multiple generalised linear models lets us test these relationships in a more satisfactory fashion. This analysis bears out the main result of the Spearman's rank correlation i.e. that most of the variability in  $k_{iso}$  is explained by soil pH. However, after controlling for the effect of pH the inclusion of nitrate availability and biomass both significantly increase the degree of variability explained (see also Table S1). This indicates that nitrate concentration does indeed control  $k_{iso}$  in natural soils. Figure 3 b shows the nature of this relationship with nitrate concentration, particularly under acidic conditions, causing  $k_{iso}$  to decrease.

19) P13L357: While the fraction of explained deviance is high, this is a small sample with n=14 and uncertainty associated with # the model could be large. What is the confidence interval of the coefficient of ln NO3-?

We agree that the sample size is small and report this model simply as the best fit to the data out of the variables considered in order to understand the influence of the treatment on the rate of exchange. Indeed, the uncertainty is large particularly at higher values of change. Please see the confidence interval provided in Figure 5.

20) P13L376–380: "Whether the potential [. . . ] remains an unresolved but key question." - Not sure what you are trying to mean with this sentence. Please clarify it.

Thanks, we have re-phrased this to make it clearer.

"Understanding why  $k_{iso}$  has the potential to be orders of magnitude greater in the field compared to values observed in incubation studies is a key question for the future. The abundance and activity of carbonic anhydrases may be reduced during the process of sieving soils and incubating them for prolonged periods in the dark. For example, the exclusion of intact roots and mycorrhizal fungi interacting within the rhizosphere might reduce kiso (Li et al., 2005). Equally the suppression of phototrophic community members by incubating mesocosms in the dark (Sauze et al., 2017) may also contribute to differences in  $k_{iso}$  between the field and incubated mesocosm experiments. Furthermore, we cannot rule out the possibility that determining  $k_{iso}$  accurately under field conditions is less reliable. For example the calculation of  $k_{iso}$  relies on determining the isotopic composition of the soil water pool in equilibrium with CO2. Given the potential for increased heterogeneity in the isotopic composition of the soil water pool in natural conditions this may make it more challenging to determine kiso robustly in the field (Jones et al., 2017)." 21) P15L425: "The absence of strong patterns with climate or land-cover in this study may well reflect the fact that the temperature and moisture conditions used are unrepresentative of field conditions especially for colder and drier sites." - Or, it could also be that soil texture and composition are the main controls.

It is true that the conditions experienced by the microbes in their natural environments can be very different from those experienced in our experiment. This would definitely be interesting to look at in the future with a different experimental and mechanistic modelling approach. However, the aim of the present study was to standardise moisture and temperature conditions to the best of our abilities and investigate how the gas exchange rates and enzyme activity of these different communities compared. Opting for this experimental design meant we were not able to attribute statistically whether differences in activity were underpinned by land-use or climate class in a way that would facilitate a simple scaling up approach, Our study indicates other soil traits such as pH have the potential to provide more reliable spatial predictions of  $k_{iso}$ . With larger databases perhaps land-use or climate patterns will begin to emerge as important large-scale drivers of soil function and predictors of soil-atmosphere gas exchange but for the moment it remains unclear as these datasets are rare in the community.

**22) P15L435: What are the "pedotransfer functions?"**

Pedotransfer functions are predictive functions used to estimate certain soil properties from more readily available data. We have altered this sentence to provide more clarity on the message we are trying to communicate:

"A significant challenge to using this statistical relationship to predict  $k_{iso}$  is underpinned by our capacity to describe the spatial and temporal variations in the important drivers of  $k_{iso}$ , namely soil pH, microbial biomass and exchangeable  $NO_3^-$ . Fortunately, a number of promising spatial databases are evolving for soil characteristics such as pH and microbial biomass likewise a number of land surface models can now estimate the spatial and temporal dynamics of the biosphere N cycle convincingly (Zaehle, 2013)."

Technical comments

23) P1L10: "gross primary production" vs. P1L25 "gross primary productivity" (emphases mine), pick one.

Thanks. L25 changed to "gross primary production".

24) P1L11: "ecosystem-scale" → "ecosystem scale"

Thanks. Corrected.

25) P1L15: Add a comma before "indicating [...]."

Thanks. Corrected.

26) P1L33: "the leaves of plants"  $\rightarrow$  "leaves". Pleonasm.

Thanks. Changed to "This is the case because leaves contain..."

27) P2L35: "causing CO2 that interacts with a leaf but is not fixed to inherit the isotopic composition of the leaf water pool" - A difficult sentence. Please clarify.

Thanks we have simplified this: "This is the case because leaves contain considerable concentrations of carbonic anhydrase that catalyses the hydration of aqueous  $CO_2$  and the exchange of oxygen isotopes between  $CO_2$  and water molecules. The rate of this exchange is rapid and causes the majority of  $CO_2$  within a leaf to inherit the isotopic composition of the leaf water pool (Gillon & Yakir, 2001)."

28) P2L44–P3L73: This paragraph has a lot to unpack. In my opinion, to bring clarity to this paragraph, you may consider splitting it into two. Describe the abiotic reaction of oxygen isotope exchange first, and then introduce the role of carbonic anhydrases in accelerating the reaction towards equilibrium. I would consider splitting the paragraph at line 62 and rearraging sentences for a clean separation.

We have rearranged and edited this section as suggested:

"The oxygen isotope composition of atmospheric  $CO_2$  is influenced by leaves and soils because oxygen isotopes are exchanged between water and  $CO_2$  through the reverse dehydration step of the reversible hydration reaction between aqueous  $CO_2$  and bicarbonate (Mills & Urey, 1940). In a closed system at chemical equilibrium,  $CO_2$  will reach isotopic equilibrium with water after some time depending on the rate of oxygen isotope exchange,  $k_{iso}$  (s-1), (Uchikawa & Zeebe, 2012). In soils the greater abundance of water molecules causes endogeneous  $CO_2$  or atmospheric  $CO_2$  that diffuses within the soil profile to inherit the  $\delta^{18}$ O of the soil water (Tans, 1998). The degree to which the  $\delta^{18}$ O of CO2 reflects a given soil water pool is determined by the residence time of dissolved CO2 and the apparent kiso (Miller et al., 1999). Longer residence times or greater kiso move the system closer to isotopic equilibrium. Resulting from the interconversion of aqueous CO2 and bicarbonate, kiso is expected to vary as a function of the combined rates of CO2 hydration, kib, and hydroxylation reactions and the pH dependent speciation of dissolved inorganic carbon (Uchikawa & Zeebe, 2012). Under acidic and neutral conditions interconversion is dominated by hydration, whilst the hydroxylation becomes important under alkaline conditions as the concentration of hydroxyl anions increases (Figure 1 a). The presence of carbonic anhydrases increases the rate of the hydration reaction, kh, and the overall rate of interconversion between CO2 and bicarbonate. However, the influence of carbonic anhydrases, for a given concentration and efficiency, is also limited by the presence of high proton concentrations under acidic conditions that inhibit de-protonation required for enzyme regeneration (Rowlett et al., 2002; Sauze et al., 2018). The kiso resulting from these reactions is dependent on the relative abundance of CO2, which is the dominant form of dissolved organic carbon under acidic conditions, to carbonic acid, bicarbonate and carbonate in the system (Figure 1 b). In alkaline conditions, the predominance of bicarbonate acts to inhibit the rate of kiso associated with the hydration reaction and limit the influence of hydroxylation (Figure 1 c).

Comprised of at least seven distinct families, the carbonic anhydrases have independently evolved in all domains of life in order to catalyse the reversible hydration of carbon dioxide (CO2) to bicarbonate described above (Jensen et al., 2019). Whilst this reaction occurs abiotically, the need for carbonic anhydrases stems from the fact that enhanced rates of hydration,  $k_h$ , are required to control the transport and availability of CO2, bicarbonate and protons in numerous metabolic processes (Smith & Ferry, 2000). Unsurprisingly given their apparent ubiquity, evidence of carbonic anhydrase activity in soils indicates the expression of these enzymes directly supports the viability of microbial communities and thus plays a role in the wider biogeochemical function of the soil environment (Li et al., 2005). The fact that  $k_{iso}$  inferred from patterns in the  $\delta^{18}$ O of CO2 fluxes observed under field (Seibt et al., 2006; Wingate et al., 2008, 2009, 2010) and laboratory conditions (Jones et al., 2017; Meredith et al., 2019; Sauze et al., 2017, 2018) can exceed uncatalysed rates by up to three orders of magnitude indicates a particular need to better understand variations in the expression of carbonic anhydrases and the controls on their activity in soil environments."

**29) P3L83: "it's" → "its"**

**Thanks. Corrected.**

30) P3L87–89: "Such an observation may result from changes in size or composition of the microbial communities involved as discussed (Sauze et al., 2017, 2018)." - This is a reiteration of P3L79–81.

**Removed.**

31) P4L95: "non-carbon"  $\rightarrow$  "non-carbonate"

Thanks. Changed to "The chemistry of other anions".

32) P5L123: "principle" → "principal"

Thanks. Corrected.

33) P5L124: "indicted" → "indicated"

Thanks. Corrected.

34) P6L171: This should be section 2.2, not 2.1.

Thanks. We have corrected section '2.1 Gas exchange measurements' to '2.2 Gas exchange measurements'.

35) P11L312–316 and P12L330–337: It is inconvenient to track which model is which. Please consider listing model diagnostics in supplementary tables.

We have added three tables to the supplement listing the relevant models discussed in the text.

Table S1: Ranking and included terms for a subset of the generalised linear models tested to predict variations in the rate of oxygen isotope exchange,  $k_{iso}$ , for the entire dataset (n = 44). Model selection was limited to a maximum of four

predictive terms and the intercept. The terms MB,  $NO_3^-$  and  $NH_4^+$  are the natural logarithms of microbial biomass and nitrate and ammonium availability. Selected terms or interactions within each model are indicated by + symbols whilst symbols indicate their omission. The interactions Campaign:pH and Campaign:MB are omitted from the table for brevity as they were not selected in any of the models shown. Model ranking was based on comparison of sample size corrected Aikake's Information Criterion (AICc) with  $\Delta$ AICc indicating the difference in AICc from the best model.  $\Delta$ AICc of 2 or more indicates real differences in model performance.

|      |           |          |    |    |                 |                   | Campaign:                    | pH: | pH:             | MB:             | NO3:              |       |
|------|-----------|----------|----|----|-----------------|-------------------|------------------------------|-----|-----------------|-----------------|-------------------|-------|
| Rank | Intercept | Campaign | pН | MB | NO 3 | $\mathbf{NH_4}^+$ | NO 3 + | MB  | NO 3 | NO 3 | $\mathbf{NH_4}^+$ | ∆AICc |
| 1    | +         | -        | +  | +  | +               | -                 | -                            | -   | +               | -               | -                 | 0.00  |
| 2    | +         | -        | +  | +  | +               | -                 | -                            | -   | -               | -               | -                 | 6.10  |
| 3    | +         | +        | +  | +  | +               | -                 | -                            | -   | -               | -               | -                 | 7.06  |
| 4    | +         | -        | +  | +  | +               | -                 | -                            | +   | -               | -               | -                 | 7.07  |
| 5    | +         | +        | +  | -  | +               | -                 | +                            | -   | -               | -               | -                 | 7.09  |
| 6    | +         | -        | +  | +  | +               | -                 | -                            | -   | -               | +               | -                 | 8.79  |
| 7    | +         | +        | +  | -  | +               | -                 | -                            | -   | -               | -               | -                 | 12.43 |
| 8    | +         | -        | -  | +  | +               | +                 | -                            | -   | -               | -               | +                 | 13.27 |
| 16   | +         | -        | +  | -  | -               | -                 | -                            | -   | -               | -               | -                 | 21.56 |
| 19   | +         | -        | -  | -  | -               | +                 | -                            | -   | -               | -               | -                 | 26.48 |
| 21   | +         | -        | -  | +  | -               | -                 | -                            | -   | -               | -               | -                 | 43.64 |
| 28   | +         | -        | -  | -  | -               | -                 | -                            | -   | -               | -               | -                 | 47.91 |
| 33   | +         | +        | -  | -  | -               | -                 | -                            | -   | -               | -               | -                 | 50.15 |
| 34   | +         | -        | -  | -  | +               | -                 | -                            | -   | -               | -               | -                 | 50.21 |

Table S2: Ranking and included terms for a subset of the generalised linear models tested to predict variations in the rate of oxygen isotope exchange,  $k_{iso}$ , for the relatively invariant soil properties of the EUR campaign dataset (n = 27). Model selection was limited to a maximum of two predictive terms and the intercept. The terms C, N and CN are soil carbon and nitrogen content and their ratio. Selected terms or interactions within each model are indicated by + symbols whilst - symbols indicate their omission. Model ranking was based on comparison of sample size corrected Aikake's Information Criterion (AICc) with  $\Delta$ AICc indicating the difference in AICc from the best model.  $\Delta$ AICc of 2 or more indicates real differences in model performance.

| Rank | Intercept | pН | Sand | Silt | Clay | С | N | CN | ΔAICc |
|------|-----------|----|------|------|------|---|---|----|-------|
| 1    | +         | +  | -    | -    | +    | - | - | -  | 0.00  |
| 2    | +         | +  | +    | -    | -    | - | - | -  | 0.57  |
| 3    | +         | +  | -    | -    | -    | + | - | -  | 1.32  |
| 4    | +         | +  | -    | -    | -    | - | - | -  | 1.85  |
| 5    | +         | +  | -    | -    | -    | - | - | +  | 1.92  |
| 6    | +         | +  | -    | +    | -    | - | - | -  | 2.46  |
| 7    | +         | +  | -    | -    | -    | - | + | -  | 4.57  |
| 8    | +         | -  | -    | -    | -    | - | + | -  | 21.26 |
| 9    | +         | -  | -    | -    | -    | - | - | -  | 22.07 |

Table S3: Ranking and included terms for a subset of the generalised linear models tested to predict variations in the change in rate of oxygen isotope exchange,  $k_{iso}$ , following ammonium nitrate addition (n = 15). Model selection was limited to a maximum of one predictive term and the intercept. The terms MB,  $NO_3^-$  and  $NH_4^+$  are differences in microbial biomass and nitrate and ammonium availability following ammonium nitrate addition whilst the prefix ln indicates the natural logarithm of these differences. Selected terms or interactions within each model are indicated by + symbols whilst - symbols indicate their omission. Model ranking was based on comparison of sample size corrected Aikake's Information Criterion (AICc) with  $\Delta$ AICc indicating the difference in AICc from the best model.  $\Delta$ AICc of 2 or more indicates real differences in model performance.

| Rank | Intercept | Campaign | pН | MB | NO 3 - | $NH_4^+$ | InMB | InNO 3 | $lnNH_4^+$ | ∆AICc |
|------|-----------|----------|----|----|------------------------------|----------|------|-------------------|------------|-------|
| 1    | +         | -        | -  | -  | -                            | -        | -    | +                 | -          | 0.00  |
| 2    | +         | -        | -  | -  | +                            | -        | -    | -                 | -          | 8.65  |
| 3    | +         | -        | +  | -  | -                            | -        | -    | -                 | -          | 13.20 |
| 4    | +         | -        | -  | -  | -                            | -        | -    | -                 | -          | 15.95 |
| 5    | +         | +        | -  | -  | -                            | -        | -    | -                 | -          | 17.38 |
| 6    | +         | -        | -  | -  | -                            | -        | +    | -                 | -          | 18.34 |
| 7    | +         | -        | -  | +  | -                            | -        | -    | -                 | -          | 18.80 |
| 8    | +         | -        | -  | -  | -                            | -        | -    | -                 | +          | 19.10 |
| 9    | +         | -        | -  | -  | -                            | +        | -    | -                 | -          | 19.21 |

36) Figure 3: It is difficult to distingush high values from low values indicated by the color bars. Try to increase the contrast.

Figure 3 has been revised to hopefully increase the contrast of the plot gradients and use a more accessible colour palette.

37) Figure S1: Remove the ocean background and other unnecessary information. Please simplify this figure to make the ecoclimatic classification more evident. Consider putting the legend outside of the figure canvas to avoid interference.

Figure S1 has been revised to mask the ocean, move the legend outside of the map area and reduce the classes to only reflect those covered by the samples obtained.

---

## Author Comment (AC3) · 13 Feb 2021

1) SOIL-2020-44 presents on an interesting research into how soil factors would affect O18 exchange between CO2 and H2O. As the 18O levels of emitted CO2 are used for estimating land-atmosphere carbon exchange it presents a well-delineated and timely topic. The experimental assessment of exchange of 18O between soil water and CO2 is by no means trivial, yet as many as 44 soils collected from a wide geographical area (Eurasia and Australia) were assessed for their potential to oxygen isotope exchange.

We would sincerely like to thank the editor for taking the time to handle the review process of this manuscript and for providing their own useful comments that have improved the manuscript. We have addressed the points raised (reproduced in blue) below.

2) The description was mostly excellent but I concur with the referee that on instances the text is lengthy. For instance the description of the statistical approach and reporting of fitted general linear models are longer than usual. However, here such an elaborate formulation appeared warranted given the clear collinearity of some of the predictor variables.

We agree that as the interpretation of the data is dependent on the statistical tests used it is important to be explicit about the steps taken.

3) Still, in particular the introduction can surely be further condensed. Also the lengthy description of the setup to administer air with CO2 with contrasting δ18O requires shortening (e.g. by just referring to or some part of it should be moved to supplementary material.

Thanks, following this good advice we have worked to streamline the revised manuscript.

4) The description of preparation of soil mesocosms could also be condensed.

Please see the previous comment.

5) Please also use a different term for available NO3- and available NH4+ (or'availability of'). Both were measured in typical 1M KCl extracts and are best referred to exchangeable NO3- and NH4+.

Thanks for this suggestion, we have modified the text accordingly to use exchangeable in place of available.

6) The entire text is filled with long complex sentences,which on their own are well crafted but do not always allow fluid reading. I would recommend the authors to read their manuscript once more and if they see fit do try to subdivide some of the longer sentences.

Thanks, following this advice we have worked to improve the writing style used in the revised manuscript.

7) I concur with the referee that some of the hypotheses require rephrasing and attention needs to be given to his/her possibly justified concern on the NH4NO3 administration's ineptitude to robustly prove that NO3-led to inhibition of carbonic anhydrase.

Please see responses to comments 1) and 4) of Reviewer 1. In brief we have rephrased the hypotheses as suggested and provided caveats about the power of the treatment experiment.

8) The interpretation of the relationship between NH4+ and kiso: On L408 following statement is made 'Notably the weak relationship between changes in kiso and NH4+availability identified in this experiment (Table 2) suggests the relationship between these variables across the untreated soils (Table 1) does indeed reflect the pH sensitivity of ammonia speciation rather than a direct causal link.'. I would agree that the link is indeed direct, but strongly doubt that NH4+/NH3 speciation is important here. Below pH 7 there is virtually no (toxic) NH3. Most soils had acidic pH so this explanation seems wrong. Instead: high NH4+ levels in upland soils rather suggest nitrification is impeded in some way. There is an obvious link between pH and NH4+ levels: at low pH, nitrification is well known to be slowed: indeed there was a strong negative correlation between both (Table 1) and so the negative correlation between NH4+ and kisomay just be indirect through their mutual relation with pH. Alternatively, higher NH4+ levels point at impeded autotrophic nitrification – a rather energetically unfavourable process and therefore sensitive to environmental constraints. Inhibited nitrification mayalso point at unfavourable conditions for other microbial processes: perhaps also pro-duction of anhydrase activity? In Table 2 the artificially elevated NH4+ levels are no longer the resultant from slow or fast pH-dependent nitrification and therefore do not display any relation with kiso. The referee also commented on this matter– please do take that comment into account.

Following this good advice we have removed the unfounded reference to the role of ammonia speciation in explaining the apparent co-correlation between $k_{iso}$, pH and exchangeable ammonium identified in the Spearman's rank analysis presented in Table 1. We maintain that the relationship between $k_{iso}$ and exchangeable $NH_4^+$ is an artefact of it's

relationship with soil pH for two reasons. Firstly, the Spearman's rank correlation between soil pH and exchangeable ammonium is strongly influenced by the co-occurrence of low exchangeable ammonium in the high pH soils that exhibit greater $k_{iso}$ (Figure 2 a) & e); groups Csa, Bsh and Cfa). Secondly, we do not find support for a role of exchangeable ammonium in explaining variability in $k_{iso}$ in the subsequent analyses (Tables S1 & S3). Please also see the response to Reviewer 1, comment 1).

9) It is unfortunate that the authors opted for an enourmous dose of NH4NO3, viz 0.7 mg NH4NO3 g-1 (L168). No doubt such a large addition of NH4+ would have led to serious soil acidification following its partial nitrification during the 1 week pre-incubation at 23◦C. And on the other hand the obtained NH4+ and NO3- levels intreated soil would not reflect environmentally realistic exchangeable N levels. That would severely limit the relevance of Fig. 5. Or is the 0.7 mg NH4NO3 g-1 a typing error? In any case in their rebuttal the authors will need to present (not necessarily in the revised manuscript) the absolute increase in NH4+ and NO3- levels alongside changes in pH brought forth by the NH4NO3 administration as based on the fractorial changes in NO3- now presented it is impossible to judge whether or not excessive amounts of N had been added. This firstly needs to be clarified.

The fertilisation rate of 0.7 mg NH4NO3 per gram of dry soil (0.25 mg of N per gram of dry soil) was adopted from Ramirez, Craine and Fierer (2012). The justification for this value (also used in Kaisermann et al., 2018) was that it approximates typically applied fertilizer loads in field studies. We have added a figure to the supplementary material (Figure S3) to provide information about the absolute values of measured parameters in both the untreated controls and treated soils and allow comparison with the wider dataset in Figure 2. The median increase in exchangeable nitrate and ammonium in the treated over untreated soils was 22 and 10 times, respectively.

[Figure]

Figure S3: Mean a) $k_{iso}$, b) pH, c) exchangeable nitrate ($NO_3^-$), d) exchangeable ammonium ($NH_4^+$) and e) microbial biomass (MB) for the untreated control and the corresponding treated soils. Dashed lines indicating the 1:1 line with points below the line representing a decrease in treated relative to untreated soils and points above the line representing an increase. Points falling along the line indicate no change.

Ramirez, K. S., Craine, J. M. and Fierer, N.: Consistent effects of nitrogen amendments on soil microbial communities and processes across biomes, Global Change Biology, 18(6), 1918–1927, https://doi.org/10.1111/j.1365-2486.2012.02639.x, 2012.

Kaisermann, A., Jones, S. P., Wohl, S., Ogée, J. and Wingate, L.: Nitrogen Fertilization Reduces the Capacity of Soils to Take up Atmospheric Carbonyl Sulphide, Soil Systems, 2(4), 62, https://doi.org/10.3390/soilsystems2040062, 2018.

10) Minor comment: Remove commas when the citation is part of the sentence throughout the entire text: Jones et al. (2017), not Jones et al., (2017).

Thanks, we have corrected this.

11) L21 suggest to omit 'future'

Thanks, done.

12) L31 'the δ18O of leaf-atmosphere CO2 exchange' reads strange, sounds clearer with-out 'exchange'

We have modified this sentence to make it clearer in general but maintain the use of the word 'exchange' as it is important to do so.

13) L 35 CO2 that interacts with a leaf – interacts seems too vague, could this be reworded?

This has been reworded in response to Reviewer 1, comment 27). It now reads: "This is because leaves contain considerable concentrations of carbonic anhydrase that catalyses the hydration of aqueous $CO_2$ and the exchange of oxygen isotopes between $CO_2$ and water molecules. The rate of this exchange is rapid and causes the majority of $CO_2$ within a leaf to inherit the isotopic composition of the leaf water pool (Gillon & Yakir, 2001)."

14) L36 'undergo considerable enrichment' of what?
Thanks, we have rephrased this section : "As leaf water pools are small and undergo considerable evaporation, the $\delta^{18}O$ of leaf water is generally enriched relative to that of the soil water (Wingate et al., 2010). Consequently, the $\delta^{18}O$ of $CO_2$ molecules that have interacted with leaf and soil water pools are also isotopically distinct from one another and can be used to constrain the contribution of soils and vegetation to the atmospheric $CO^{18}O$ mass balance  (Francey & Tans, 1987)."

15) L40 'alters the δ18O of atmospheric CO2' was not really clear to me. Perhaps write 'alters the δ18O of CO2 in soil vs. atmospheric CO2'?
We have simplified this sentence "Currently, the abundance and activity of carbonic anhydrases in soil is poorly understood complicating our abilities to predict the influence of soil communities on the oxygen isotope composition of atmospheric $CO_2$."

16) L51 is 'of soil atmospheric CO2' not better?

This section has been restructured in response to Reviewer 1, comment 28).

17) L56 'abiotically invades' sounds awkward, perhaps write 'diffuses into the soil porenetwork'

We specifically used the term "invade" as we were referring to the invasion flux of $CO_2$ (also known as the piston velocity of the overlying air-column). However, to simplify we have changed the text to "that diffuses within the soil profile"

18) L65 delete 'true', is confusing here

Thanks, deleted.

19) L99 better 'agricultural soils'?

Thanks, corrected 'agriculture' to 'agricultural'

20) L166 'an additional three replica incubations' correct English?

We have re-written this paragraph to improve the clarity of our methodology.

"An $NH_4NO_3$ addition experiment was also conducted in both campaigns. This involved the preparation of three additional replicated incubations as described above, for nine of the EUR sites and five of the AUS sites. Prior to the pre-incubation step, 0.7 mg of $NH_4NO_3$ g dry soil$^{-1}$ was dissolved in water and used to adjust the water content of three of the replicated incubations. These were then incubated with the three other 'control' incubations in the same control chamber."

21) 2.1 Not clear if soils were kept cool during transport

We have clarified that EUR samples were transported at ambient temperatures and that AUS samples were returned to the laboratory on the day of sampling to minimise the time the samples spent without definite temperature control.

22) L273 remove 'presumably reflecting the pH dependency of NH4+ and ammonia speciation' this does not belong in the M&M section

Thanks, this has been deleted.

23) L293-294 do not seem entirely accurate: there seems to be a higher kiso for the Csa group.

It's true there are two sites associated with the Csa group that have high values above the 0.75 quantiles (i.e. the upper hinge of the box). However, these values are lower than the unique case for Bsh and an individual from the Cwa group. Furthermore the median of the Csa group clearly overlaps with the 0.50 to 0.75 quantile of both the Cfa and Cwa groups. On balance we feel that this does not support the conclusion that there is a robust pattern associated with these climate and landcover classifications. We do not attempt to test this statistically simply because despite our efforts the number of replicates for land-cover and climate still remains relatively low and their distribution is unbalanced.

24) L370-375 is quite unclear: are the 0.04 to 13 s-1 field values derived from your labmeasured 0. To 0.4 s-1 kiso estimates? + which older studies. The whole part starts quite sudden. In fact for the sake of clarity the apparent issue of comparing lab and field estimates is perhaps best omitted from the paper entirely – also further on.

We feel it is important to include the comparison of $k_{iso}$ observed in this study with those from the literature. This type of information has not typically been presented in recent publications in part because it is not always easy to compare reported values on an equal basis. However, by attempting to do so we do observe that there are potentially significant discrepancies that should not be ignored. We have adjusted and rephrased this paragraph to hopefully make it clearer:
"The observations of this study, with a median of 0.07 s$^{-1}$, are in good agreement with a number of previous studies (Jones et al., 2017; Sauze et al., 2018, 2017) that estimated $k_{iso}$ values ranging from 0.03 to 0.15 s$^{-1}$ for sieved soils incubated in the dark. However, these rates are somewhat lower than those reported by Meredith et al. (2019) for dark incubated soils with a median and range of 0.46 s$^{-1}$ and 0.08 to 0.88 s$^{-1}$, respectively. These greater $k_{iso}$ values reported by Meredith et al. (2019) are more comparable to those, ranging from 0.01 to 0.75 s$^{-1}$, reported by Sauze et al. (2017) for soils with well developed algal communities. Direct comparison of our estimates $k_{iso}$ with those observed in the field is challenging because these older studies (Seibt et al., 2006; Wingate et al., 2008, 2009, 2010) estimated soil carbonic anhydrase activity as a range of enhancement factors over a temperature sensitive uncatalysed rate of hydration . However, using the mid-point of the enhancement factors and soil temperatures reported by Wingate et al. (2009), we estimate that $k_{iso}$ varied between 0.04 and 13 s$^{-1}$ with a median of 0.31 s$^{-1}$ across the seven ecosystems considered in their analysis . Understanding why $k_{iso}$ has the potential to be orders of magnitude greater in the field compared to values observed in lab incubation studies is a key question for the future. The abundance and activity of carbonic anhydrases may be reduced during the process of sieving soils and incubating them for prolonged periods in the dark. For example, the exclusion of intact roots and mycorrhizal fungi interacting within the rhizosphere might reduce $k_{iso}$ (Li et al., 2005). Equally the suppression of phototrophic community members by incubating mesocosms in the dark (Sauze et al., 2017) may also contribute to differences in $k_{iso}$ between the field and incubated mesocosm experiments. Furthermore, we cannot rule out the possibility that determining $k_{iso}$ accurately under field conditions is less reliable. For example the calculation of $k_{iso}$ relies on determining the isotopic composition of the soil water pool in equilibrium with $CO_2$. Given the potential for increased heterogeneity in the isotopic composition of the soil water pool in natural conditions this may make it more challenging to determine $k_{iso}$ robustly in the field (Jones et al., 2017)."

25) L431 'between untreated and treated' is not a very clear phrasing – spell out betterwhat you are referring at.
We have rephrased this to hopefully make the meaning clearer:

"Indeed, the ability of this model to reasonably predict fractional changes in $k_{iso}$ between untreated control soils, that were used to build the model, and their fertiliser treated counterparts, that were not used to 'train' the model selection process, is encouraging (Figure 4 b)."

26) L434 'A significant challenge to using this relationship to predict kiso is likely the avail-ability of suitable pedotransfer functions,particularly for NO3−availability and microbial biomass, to estimate patterns in the proposed drivers (Van Looy et al., 2017).' Is an understatement. It will be impossible to predict the ephemeral soil NO3- levels with asimple pedotransfer function. The authors best refer to use soil N models to predictNO3 levels and use these as inputs into Eq 6.

We have rephrased this section in response to Reviewer 1, comment 22) and refer to the need for dynamic models in predicting these parameters: "A significant challenge to using this statistical relationship to predict $k_{iso}$ is underpinned by our capacity to describe the spatial and temporal variations in the important drivers of $k_{iso}$, namely soil pH, microbial biomass and exchangeable $NO_3^-$. Fortunately, a number of promising spatial databases are evolving for soil characteristics such as pH and microbial biomass likewise a number of land surface models can now estimate the spatial and temporal dynamics of the biosphere N cycle convincingly (Zaehle, 2013)."

Zaehle S. 2013 Terrestrial nitrogen–carbon cycle interactions at the global scale. Phil Trans R Soc B 368: 20130125. http://dx.doi.org/10.1098/rstb.2013.0125

---

## Author Response (AR2)

**Author's response to editors decision (in blue)**

**"Topical Editor Decision: Publish subject to minor revisions**
Comments raised by the referee, public and AE have all been properly addressed. The in depth reply on some of the raised points is much appreciated. The most crucial issues: our doubts on the relation between NO3 and kiso, and effect of NH4+ on kiso have been resolved. Also foreseen improvements of the readability of the text are expected to be sufficient. We welcome submission of a final revised MS that implements the suggested changes by the authors."

Following the editors decision of we have implemented the changes outlined in our intial responses to the online discussion. We have also editted the text and re-written the introduction following the good advice of both the editor and the reviewer to improve the readability.  We would like to thank Steven Sleutel, Gerbrand Koren and Reviewer 1 for their time and input.

**Author's response to review comments**

We have listed our responses to the reviewer comments (reproduced in blue) along with the final changes made below. In some cases the exact content of our responses to comments about writing style or clarity now differ to those made during the dicussion phase because we editted and re-written some sections.

**EC1**

1) SOIL-2020-44 presents on an interesting research into how soil factors would affect O18 exchange between CO2 and H2O. As the 18O levels of emitted CO2 are used for estimating land-atmosphere carbon exchange it presents a well-delineated and timely topic. The experimental assessment of exchange of 18O between soil water and CO2 is by no means trivial, yet as many as 44 soils collected from a wide geographical area (Eurasia and Australia) were assessed for their potential to oxygen isotope exchange.

We would sincerely like to thank the editor for taking the time to handle the review process of this manuscript and for providing their own useful comments that have improved the manuscript. We have addressed the points raised (reproduced in blue) below.

2) The description was mostly excellent but I concur with the referee that on instances the text is lengthy. For instance the description of the statistical approach and reporting of fitted general linear models are longer than usual. However, here such an elaborate formulation appeared warranted given the clear collinearity of some of the predictor variables.

We agree that as the interpretation of the data is dependent on the statistical tests used it is important to be explicit about the steps taken.

3) Still, in particular the introduction can surely be further condensed. Also the lengthy description of the setup to administer air with CO2 with contrasting δ18O requires shortening (e.g. by just referring to or some part of it should be moved to supplementary material.

We have rewritten the introduction and editted the text elsewhere to remove unnesserary information.

4) The description of preparation of soil mesocosms could also be condensed.

Please see the previous comment.

5) Please also use a different term for available NO3- and available NH4+ (or'availability of'). Both were measured in typical 1M KCl extracts and are best referred to exchangeable NO3- and NH4+.

Thanks for this suggestion, we have modified the text accordingly to use exchangeable in place of available.

6) The entire text is filled with long complex sentences,which on their own are well crafted but do not always allow fluid reading. I would recommend the authors to read their manuscript once more and if they see fit do try to subdivide some of the longer sentences.

We have worked to improve the writing style throughout the abstract and main text (see also comment 3 and 4).

7) I concur with the referee that some of the hypotheses require rephrasing and attention needs to be given to his/her possibly justified concern on the NH4NO3 administration's ineptitude to robustly prove that NO3-led to inhibition of carbonic anhydrase.

Please see responses to comments 1) and 4) of Reviewer 1. In brief we have rephrased the hypotheses as suggested and provided caveats about the power of the treatment experiment.

8) The interpretation of the relationship between NH4+ and kiso: On L408 following statement is made 'Notably the weak relationship between changes in kiso and NH4+availability identified in this experiment (Table 2) suggests the relationship between these variables across the untreated soils (Table 1) does indeed reflect the pH sensitivity of ammonia speciation rather than a direct causal link.'. I would agree that the link is indeed direct, but strongly doubt that NH4+/NH3 speciation is important here. Below pH 7 there is virtually no (toxic) NH3. Most soils had acidic pH so this explanation seems wrong. Instead: high NH4+ levels in upland soils rather suggest nitrification is impeded in some way. There is an obvious link between pH and NH4+ levels: at low pH, nitrification is well known to be slowed: indeed there was a strong negative correlation between both (Table 1) and so the negative correlation between NH4+ and kisomay just be indirect through their mutual relation with pH. Alternatively, higher NH4+ levels point at impeded autotrophic nitrification – a rather energetically unfavourable process and therefore sensitive to environmental constraints. Inhibited nitrification mayalso point at unfavourable conditions for other microbial processes: perhaps also pro-duction of anhydrase activity? In Table 2 the artificially elevated NH4+ levels are no longer the resultant from slow or fast pH-dependent nitrification and therefore do not display any relation with kiso. The referee also commented on this matter– please do take that comment into account.

Following this good advice we have removed the unfounded reference to the role of ammonia speciation in explaining the apparent co-correlation between $k_{iso}$, pH and exchangeable ammonium identified in the Spearman's rank analysis presented in Table 1. We maintain that the relationship between $k_{iso}$ and exchangeable $NH_4^+$ is an artefact of it's relationship with soil pH for two reasons. Firstly, the Spearman's rank correlation between soil pH and exchangeable ammonium is strongly influenced by the co-occurrance of low exchangeable ammonium in the high pH soils that exhibit greater $k_{iso}$ (Figure 2 a) & e); groups Csa, Bsh and Cfa). Secondly, we do not find support for a role of exchangeable ammonium in explaining variability in $k_{iso}$ in the subsequent analyses (Tables S1 & S3). Please also see the response to Reviewer 1, comment 1).

9) It is unfortunate that the authors opted for an enourmous dose of NH4NO3, viz 0.7 mg NH4NO3 g-1 (L168). No doubt such a large addition of NH4+ would have led to serious soil acidification following its partial nitrification during the 1 week pre-incubation at 23◦C. And on the other hand the obtained NH4+ and NO3- levels intreated soil would not reflect environmentally realistic exchangeable N levels. That would severely limit the relevance of Fig. 5. Or is the 0.7 mg NH4NO3 g-1 a typing error? In any case in their rebuttal the authors will need to present (not necessarily in the revised manuscript) the absolute increase in NH4+ and NO3- levels alongside changes in pH brought forth by the NH4NO3 administration as based on the fractorial changes in NO3- now presented it is impossible to judge whether or not excessive amounts of N had been added. This firstly needs to be clarified.

The fertilisation rate of 0.7 mg NH4NO3 per gram of dry soil (0.25 mg of N per gram of dry soil) was adopted from Ramirez, Craine and Fierer (2012). The justification for this value (also used in Kaisermann et al., 2018) was that it approximates typically applied fertilizer loads in field studies. We have added a figure to the supplementary material (Figure S3) to provide information about the absolute values of measured parameters in both the untreated controls and treated soils and allow comparison with the wider dataset in Figure 2. The median increase in exchangeable nitrate and ammonium in the treated over untreated soils was 22 and 10 times, respectively.

[Figure]

*Figure S3: Mean a) $k_{iso}$, b) pH, c) exchangeable nitrate ($NO_3^-$), d) exchangeable ammonium ($NH_4^+$) and e) microbial biomass (MB) for the untreated control and the corresponding treated soils. Dashed lines indicating the 1:1 line with points below the line representing a decrease in treated relative to untreated soils and points above the line representing an increase. Points falling along the line indicate no change.*

Ramirez, K. S., Craine, J. M. and Fierer, N.: Consistent effects of nitrogen amendments on soil microbial communities and processes across biomes, Global Change Biology, 18(6), 1918–1927, https://doi.org/10.1111/j.1365-2486.2012.02639.x, 2012.

Kaisermann, A., Jones, S. P., Wohl, S., Ogée, J. and Wingate, L.: Nitrogen Fertilization Reduces the Capacity of Soils to Take up Atmospheric Carbonyl Sulphide, Soil Systems, 2(4), 62, https://doi.org/10.3390/soilsystems2040062, 2018.

10) Minor comment: Remove commas when the citation is part of the sentence throughout the entire text: Jones et al. (2017), not Jones et al., (2017).

Corrected.

11) L21 suggest to omit 'future'

This has been removed as part of the re-write and editting implemented in response to comment 3.

12) L31 'the δ18O of leaf-atmosphere CO2 exchange' reads strange, sounds clearer with-out 'exchange'

We have modified this as part of our response to comment 3) but we maintain the use of the word "exchange" in reference to leaf-atmosphere and soil-atmosphere exchanges through-out the text as it is important to do so.

13) L 35 CO2 that interacts with a leaf – interacts seems too vague, could this be reworded?

This section has been rewritten as part of our response to comment 3).

14) L36 'undergo considerable enrichment' of what?
This section has been rewritten as part of our response to comment 3).

15) L40 'alters the δ18O of atmospheric CO2' was not really clear to me. Perhaps write 'alters the δ18O of CO2 in soil vs. atmospheric CO2'?
This section has been rewritten as part of our response to comment 3).

16) L51 is 'of soil atmospheric CO2' not better?

This section has been rewritten as part of our response to comment 3).

17) L56 'abiotically invades' sounds awkward, perhaps write 'diffuses into the soil porenetwork'

We specifically used the term "invade" as we were referring to the invasion flux of $CO_2$ (also known as the piston velocity of the overlying air-column; L52).

18) L65 delete 'true', is confusing here

This section has been rewritten as part of our response to comment 3).

19) L99 better 'agricultural soils'?

'agriculture' changed to 'agricultural' (L!03)

20) L166 'an additional three replica incubations' correct English?

We have re-written this paragraph to improve the clarity of our methodology.

L169 – L174: "An NH4NO3 addition experiment was also conducted in both campaigns. This involved the preparation of three additional 170 replicated incubations as described above, for nine of the EUR sites and five of the AUS sites. Prior to the pre-incubation step, 0.7 mg of NH 4 NO 3 g dry soil −1 was dissolved in water and used to adjust the water content of these additional replicate incubations . These were then incubated alongside the three other 'control' incubations prepared as part of the spatial survery described above."

21) 2.1 Not clear if soils were kept cool during transport

We have clarified that EUR samples were transported at ambient temperatures (L135) and that AUS samples were returned to the laboratory on the day of sampling (L154).

22) L273 remove 'presumably reflecting the pH dependency of NH4+ and ammonia speciation' this does not belong in the M&M section

Thanks, this has been deleted.

23) L293-294 do not seem entirely accurate: there seems to be a higher kiso for the Csa group.

It's true there are two sites associated with the Csa group that have high values above the 0.75 quantiles (i.e. the upper hinge of the box). However, these values are lower than the unique case for Bsh and an individual from the Cwa group. Furthermore the median of the Csa group clearly overlaps with the 0.50 to 0.75 quantile of both the Cfa and Cwa groups. On balance we feel that this does not support the conclusion that there is a robust pattern associated with these climate and landcover classifications. We do not attempt to test this statistically simply because despite our efforts the number of replicates for land-cover and climate still remains relatively low and their distribution is unbalanced.

24) L370-375 is quite unclear: are the 0.04 to 13 s-1 field values derived from your labmeasured 0. To 0.4 s-1 kiso estimates? + which older studies. The whole part starts quite sudden. In fact for the sake of clarity the apparent issue of comparing lab and field estimates is perhaps best omitted from the paper entirely – also further on.

We feel it is important to include the comparison of $k_{iso}$ observed in this study with those from the literature. This type of information has not typically been presented in recent publications in part because it is not always easy to compare reported values on an equal basis. However, by attempting to do so we do observe that there are potentially significant discrepancies that should not be ignored. We have adjusted and rephrased this paragraph.

L365 – 388: *"This study aimed to reveal the drivers of variations in the oxygen isotope exchange rate, $k_{iso}$, to make it possible to predict the influence of different soil characteristics on the $\delta^{18}O$ of atmospheric $CO_2$ and improve our understanding of soil CA activity. To do so, controlled incubation experiments were conducted to estimate $k_{iso}$ from soils collected across western Eurasia and northeastern Australia. Estimates of $k_{iso}$ for untreated soils in this study ranged from 0.01 to 0.4 $s^{-1}$ (Fig. 2 a). In all cases these rates exceeded theoretical uncatalysed rates (from 0.00008 to 0.008 $s^{-1}$ depending on soil pH, Uchikawa & Zeebe, 2012), indicating the presence of active CAs. The median $k_{iso}$ of 0.07 $s^{-1}$ reported here is in the range of previously published values for sieved soils incubated in the dark (between 0.03 and 0.15 $s^{-1}$, Jones et al., 2017; Sauze et al., 2018, 2017) but lower than those reported by Meredith et al. (2019) with a median and range of 0.46 $s^{-1}$ and 0.08 to 0.88 $s^{-1}$, respectively. These greater $k_{iso}$ values reported by Meredith et al. (2019) are more comparable to values (between 0.01 to 0.75 $s^{-1}$) reported by Sauze et al. (2017) for soils with well-developed phototroph communities. Direct comparison of our estimates of $k_{iso}$ with those observed in the field is challenging because these older studies (Seibt et al., 2006; Wingate et al., 2008, 2009, 2010) estimated soil CA activity as a range of enhancement factors over a temperature sensitive uncatalysed rate of hydration. However, using the mid-*

*point of the enhancement factors and soil temperatures reported by Wingate et al. (2009), we estimate that $k_{iso}$ varied between 0.04 and 13 $s^{-1}$ with a median of 0.31 $s^{-1}$ across the seven ecosystems considered in their analysis. Understanding why $k_{iso}$ can be orders of magnitude greater in the field compared to values observed in laboratory incubations is a key question for further studies. Potentially, the abundance and activity of CAs may be reduced during the process of sieving soils and incubating them for prolonged periods in the dark. For example, the exclusion of intact roots and mycorrhizal fungi interacting within the rhizosphere might reduce $k_{iso}$ (Li et al., 2005). Equally the suppression of phototrophic community members by incubating soils in the dark (Sauze et al., 2017) may also contribute to differences in $k_{iso}$ between the field and such experiments. Furthermore, we cannot exclude the possibility that determining $k_{iso}$ accurately under field conditions is less reliable. For example, the calculation of $k_{iso}$ relies on determining the $\delta^{18}O$ of the soil water pool in equilibrium with $CO_2$. Given the potential for increased heterogeneity in the soil water pool in natural conditions this may make it more challenging to determine $k_{iso}$ robustly in the field (Jones et al., 2017). "*

25) L431 'between untreated and treated' is not a very clear phrasing – spell out betterwhat you are referring at.
We have rephrased this to hopefully make the meaning clearer.

L442-444: *"Indeed, the ability of this model to reasonably predict fractional changes in $k_{iso}$ between untreated control soils, that were used to build the model, and their fertiliser treated counterparts, that were not used to 'train' the model selection process, is encouraging (Fig. 4 b)."*

26) L434 'A significant challenge to using this relationship to predict kiso is likely the avail-ability of suitable pedotransfer functions,particularly for NO3−availability and microbial biomass, to estimate patterns in the proposed drivers (Van Looy et al., 2017).' Is an understatement. It will be impossible to predict the ephemeral soil NO3- levels with asimple pedotransfer function. The authors best refer to use soil N models to predictNO3 levels and use these as inputs into Eq 6.

We have rephrased this section in response to Reviewer 1, comment 22) and the point raised here.

L445 – L460: *"A significant challenge to using this statistical relationship to predict $k_{iso}$ is underpinned by our capacity to describe the spatial and temporal variations in the important drivers of $k_{iso}$, namely soil pH, microbial biomass and exchangeable $NO_3^-$. For this reason we also considered whether more readily available parameters such as soil texture, carbon content and nitrogen content might provide an alternative basis for empirical predictions of $k_{iso}$ (Van Looy et al., 2017). However, relationships between these variables and $k_{iso}$ were relatively weak and could only explain a marginal amount of the observed variability. Fortunately, a number of promising spatial databases are evolving for soil characteristics such as pH and microbial biomass (Serna-Chavez et al., 2013; Slesserev et al., 2016). Likewise a number of land surface models can now estimate the spatial and temporal dynamics of the biosphere nitrogen cycle convincingly (Zaehle, 2013). Predictions of soil nutrient dynamics will likely depend on the use of such advanced soil nitrogen cycle models. Given the interaction between soil pH and exchangeable $NO_3^-$ (Fig. 3 a & b), the absence of such data may not seriously compromise predictions for fertilised agricultural soils as typically they are not strongly acidic. However, accurately predicting natural spatial and seasonal variability and the influence of future changes in atmospheric $NO_3^-$ deposition (DeForest et al., 2004) may be more problematic. Nonetheless, the data reported in this study now lay the foundations for an empirical approach to predicting $k_{iso}$ for a wide range of soils using readily available maps of key soil traits. This represents an important breakthrough in predicting how variations in soil community CA activity impacts the $\delta^{18}O$ of atmospheric $CO_2$."*

**RC1**

1) My main concern is with the interpretation of the results of ammonium nitrate (NH4NO3) treatment. The authors attributed the decrease of kiso following NH4NO3 addition to the inhibition of carbonic anhydrase caused by NO3-. However, other possible mechanisms, namely, inhibition through increased ammonium content or decreased pH cannot be ruled out by the experimental design, nor by the statistical analysis that follows.

In essence, NH4NO3 addition may affect kiso through these causal pathways:
• NH4NO3 addition → [NH4+] increase → kiso decrease
• NH4NO3 addition → [NH4+] increase → pH decrease → kiso decrease
• NH4NO3 addition → [NO3-] increase → kiso decrease

To accept Hypothesis 3, the authors must show evidence that after controlling for all confounding variables, including pH and [NH4+], there is still a robust decrease of kiso with the increase of [NO3-]. Given the absence of a randomized design and the small sample size (n=14) for NH4NO3 addition treatment, it is difficult to identify [NO3-] as the unique cause for carbonic anhydrase inhibition. One possible solution could be to treat pH and [NH4+] as instrumental variables, but this would require them to show strong correlation with [NO3-]. The best way would be to separate different causes through experimental design.

We agree that the experimental design of the ammonium nitrate treatment is not sufficient to fully tease apart the combined systematic effects (i.e. increased nitrate and ammonium availability and decreased soil pH) of the treatment on $k_{iso}$. As the reviewer states, a more extensive controlled factorial experiment would be required to achieve this. However, the results of this experiment (Section 3.2; Figure 5; Table S3), that show changes in $k_{iso}$ are most strongly linked to changes in nitrate availability (pathway 3 above) and to a lesser degree soil pH (pathway 2 above) but do not appear related to changes in ammonium availability (pathway 1 above), are still informative to the interpretation of the wider study. Across the untreated soils we clearly identify soil pH, nitrate availability and microbial biomass as explaining variations in $k_{iso}$ (Section 3.1; Figure 3; Table S1). Agreement between the results of both these analyses helps reinforce the importance of pH and nitrate (pathways 2 and 3) but not, directly at least, a role for ammonium (pathway 1). We have adjusted the text in the abstract and Section 4 to acknowledge that limitations of the experimental treatment prevent the definite conclusion that only nitrate, and not some combination of effects, influences the decrease in $k_{iso}$ observed following the fertilisation treatment.

L25 - L26: *"This effect appears to be supported by a supplementary ammonium nitrate fertilisation experiment conducted on a subset of the soils"*

L426 – L430: *"It is important to note that whilst the relationship between the changes in $k_{iso}$ and exchangeable $NO_3^-$ are supported by observations from the untreated dataset, the experimental design used in this addition experiment is not sufficient to fully test the influence of the combined changes in soil pH, exchangeable $NO_3^-$, exchangeable $NH_4^+$ and microbial biomass on $k_{iso}$. Further controlled, factorial experiments are needed for this purpose."*

2) A minor concern I have is that this study was partially motivated by the use of δ18O of CO2 to estimate terrestrial photosynthesis. While the validity of this method has been demonstrated at the global scale by Welp et al. (2011), I would caution that it is un-clear whether the current in-situ observational network would provide sufficient data to resolve regional-scale photosynthesis. Nevertheless, in my opinion, soil–atmosphere CO2 isotope exchange is an interesting topic for its own sake, regardless of whether δ18O-CO2 can provide constraints on terrestrial photosynthesis with accuracy and spatio-temporal resolution as high as those of other photosynthetic tracers in vogue (e.g., solar-induced chlorophyll fluorescence).

We agree with the reviewer that the current in-situ observational network of $\delta^{18}O$ in atmospheric $CO_2$ is rather coarse, at least compared to the network for total $CO_2$ mixing ratio. However, there are still more than 50 atmospheric stations measuring $\delta^{18}O$ in $CO_2$, spread across all latitudes and continents, with some of them covering several decades of measurements. The extremely large north-south gradient of $\delta^{18}O$ in $CO_2$ and its seasonal and interannual dynamics brings unique information on the seasonality and inter-annual variability of the northern hemisphere $CO_2$ sink, which is the strongest land carbon sink at the global scale and with the largest long-term trend (Ciais et al. 2019). Currently, this information is obscured by the lack of understanding of how soil dwelling organisms (and their carbonic anhydrase activity) affect this signal (Wingate et al. 2009). This study presents the largest soil dataset ever gathered on soil $\delta^{18}O$-$CO_2$ exchange. The in-depth analysis of the drivers of soil carbonic anhydrase activity that this study brings also serves as an important stepping-stone to study other emerging tracers of the carbon cycle including the $\Delta^{17}O$ anomaly in $CO_2$ (Koren et al. 2019) and COS (Campbell et al., 2017). For all these reasons, it would seem awkward not to mention the implication this study will have in the future development of independent tracers to study the global carbon cycle. We also agree that solar-induced chlorophyll fluorescence (SIF), another independent proxy of photosynthesis, has the

advantage of being detected from space, conferring a global coverage. However the relationship between SIF detected from space and land photosynthesis is still not well understood, notably in disentangling structural and physiological factors. For this reason, SIF is most interesting at very high spatial resolution, which is only possible since the late 2010s, with satellite instruments like TROPOMI launched in 2017 or FLEX that will be launched in 2022. Here we do not pretend that $\delta^{18}O$-$CO_2$ is a more powerful tracer compared to other tracers of global photosynthesis (i.e. SIF or COS), but we are convinced that $\delta^{18}O$-$CO_2$ contains unique independent and historical information, strongly linked to the global water and carbon cycle, that cannot be discarded. This study, with its extensive survey of soil types and biomes, addresses one of the key knowledge gaps that currently prevent the routine use of $\delta^{18}O$-$CO_2$ as a global carbon tracer, and should motivate the community to reconsider this independent tracer in global climate models, thus constraining our understanding of variability in the northern hemisphere land carbon sink. Hopefully, this may also stimulate the development of a denser observational network of $\delta^{18}O$ in $CO_2$, which is now possible with the next generation of laser-based $CO_2$ isotope analysers.

Campbell, J. E., Berry, J. A., Seibt, U., Smith, S. J., Montzka, S. A., Launois, T., Belviso, S., Bopp, L. and Laine, M.: Large historical growth in global terrestrial gross primary production, Nature, 544(7648), 84, https://doi.org/10.1038/nature22030, 2017.

Ciais P, Tan J, Wang X et al. (2019) Five decades of northern land carbon uptake revealed by the inter-hemispheric CO2 gradient. Nature, 568, 221–225.

Koren G, Schneider L, van der Velde IR et al. (2019) Global 3-D Simulations of the Triple Oxygen Isotope Signature Δ17O in Atmospheric CO2. Journal of Geophysical Research-Atmospheres, 127, 73.

Wingate L, Ogee J, Cuntz M et al. (2009) The impact of soil microorganisms on the global budget of δ18O in atmospheric CO2. Proceedings of the National Academy of Sciences of the United States of America, 106, 22411–22415.

3) The writing needs more clarity and conciseness. As a rule of thumb, try not to make sentences more complicated than the ideas they convey. In a paragraph, stick to one point and avoid switching topics or walking back and forth. For example, much of the discussion had the main points hidden in the middle of a paragraph and could use some restructuring. Break long paragraphs if necessary.

Following this comment and similar comments made in EC1 we have re-wrriten the introduction and edited both abtracted and text to improve the readability of the manuscript.

4) The hypotheses need to be accurately framed. Hypothesis 2 is a complicated statement, and the only part testable based on your experiments is that kiso increases with soil pH. The rest of Hypothesis 2 describes possible mechanisms and they cannot be answered by your experiments. In Hypothesis 3, you can only test whether kiso increases with [NO3-], but not whether [NO3-] binds carbonic anhydrases or how it inhibits carbonic anhydrase. These two hypotheses should be precisely worded as testable hypotheses. The hypotheses you actually tested were stated in P14L382–383, so why not simplify them just like that?

We have simplified our hypotheses to following this advice.

L112 – L 115: *"Based on the potential controls on $k_{iso}$ presented above we tested three specific, non-exclusive, hypotheses; 1) $k_{iso}$ increases as microbial biomass increases (H1), 2) $k_{iso}$ increases as soil pH increases (H2), and 3) $k_{iso}$ decreases as the presence of $NO_3^-$ increases (H3)."*

5) Finally, I encourage the authors to make the data sets publicly available in a data repository. This would make the study more easily discoverable and facilitate data reuse in future studies, for example, comparison across sites and parameterization of related soil processes in a land biosphere model.

We have archived the data with PANGAEA.

L662 – L664: *"The data produced in this study have been achived with PANGAEA (https://doi.org/10.1594/PANGAEA.928394). The data may also be requested from the corresponding author by email."*

Specific comments

6) P1L13: "The expression and activity of carbonic anhydrase [. . . ]" - You may need to tell the reader that carbonic anhydrase regulates the hydration of CO2 in soil pore-space water before you mention that it drives kiso.

We have rephrased this sentence as suggested.

L17 – L 19: *"As the carbonic anhydrases (CAs) group of enzymes enhances the rate of $CO_2$ hydration within the water-filled pore spaces of soils it is important to develop understanding of how environmental drivers can impact $k_{iso}$ through changes in their activity. "*

7) P1L19–20: "[. . . ] potentially reflecting the direct or indirect inhibition of carbonic anhydrases" - Is there a way to tell which mechanism is more likely?

To distinguish whether the impact of nitrate is direct or indirect an integrated study looking into changes in the concentration of carbonic anhydrase protein and the abundance of carbonic anhydrase transcripts alongside measurements of $k_{iso}$ would be required. Additionally it would also be important to do some detailed protein studies that show the physical interaction of nitrate with the carbonic anhydrase protein and develop a method that could quantify the binding efficiency of nitrate to carbonic anhydrase for a few of the dominant soil carbonic anhydrases e.g. the beta-CA class. Collectively these different experiments would help us tease apart the direct and indirect effects of nitrate on carbonic anhydrase in soils.

8) P2L31: "because the δ18O of leaf–atmosphere CO2 exchange tends to be enriched [. . . ]" - More precisely, this is because leaf preferentially uses lighter isotopologues of CO2, which diffuse faster than heavier ones. See Farquhar et al. (1993) Nature (https://doi.org/10.1038/363439a0).

This section has now been removed as part of the re-write of the introduction. However, diffusion is not the only reason. We agree that the oxygen isotope composition of leaf-atmosphere $CO_2$ exchange is partly explained by fractionation during diffusion, but not only by this. The isotopic exchange between $CO_2$ and water is also very important (Farquhar et al. 1993). In contrast the influence of oxygen isotope fractionation during other steps of fixation (e.g. carboxylation) is limited because carbonic anhydrase concentrations are sufficiently high enough for the isotopic equilibration between $CO_2$ and water to be extremely rapid (Ogée et al. 2018). By analogy to $^{13}C$ fractionation during photosynthesis, Farquhar et al. (1993) described the leaf as consuming isotopically lighter $CO_2$ in terms of $^{18}O$, thereby leaving behind $CO_2$ enriched in $^{18}O$ in the intercellular air space to diffuse back to the atmosphere. However, the analogy works because the $CO_2$ inside the leaf equilibrates its oxygen isotopes with evaporatively enriched leaf water. Thus, the mechanism is very different than for $^{13}C$, and primarily driven by leaf water isotopic composition and secondarily by diffusion.

Farquhar, G. D., Lloyd, J., Taylor, J. A., Flanagan, L. B., Syvertsen, J. P., Hubick, K. T., Wong, S. C. and Ehleringer, J. R. (1993) Vegetation effects on the isotope composition of oxygen in atmospheric CO2, Nature, 363(6428), 439–443, doi:10.1038/363439a0.

Ogée J, Wingate L, Genty B (2018) Estimating mesophyll conductance from measurements of C18OO photosynthetic discrimination and carbonic anhydrase activity. Plant Physiol., 178, 728–752.

9) P2L44: "Comprising at least six distinct families, [. . . ]" - There are seven now, with the newly discovered ι-CA in phytoplanktons. See Jensen et al. (2019) ISME J (https://doi.org/10.1038/s41396-019-0426-8).

The introduction and references now include this information.
L60 – L62: "Currently, at least seven distinct CA gene families have been identified, with each catalysing the reversible hydration of $CO_2$ to bicarbonate (Jensen et al., 2019)."

L540 – L545: "Jensen, E. L., Clement, R., Kosta, A., Maberly, S. C. and Gontero, B.: A new widespread subclass of carbonic anhydrase in marine phytoplankton, The ISME Journal, 13(8), 2094–2106, doi:10.1038/s41396-019-0426-8, 2019."

10) P3L81–82: "Whilst the sensitivity of soil kiso to the presence of specific functional groups, like phototrophs which employ carbonic anhydrases in their carbon concentration mechanisms [. . . ]" - Are phototrophs abundant in soil microbial communities?

In a review of the literature Wingate et al., 2009 estimated that soil algal populations of between $10^3$ - $10^6$ per gram of soil are typically present in most soils. If cyanobacteria are further included, phototrophs can indeed form an important

part of the soil microbial community under many conditions (Muriel Bristol Roach, 1927; Seppey et al., 2017). This may be either as superficial crusts or within the near surface. Whilst they are likely to be less ubiquitous than fungi and bacteria, the possibility of specialised, carbonic anhydrase dependent, carbon concentration mechanisms might suggest their presence could have a disproportionately strong influence on $k_{iso}$. In a previous study looking at the role of phototrophs on carbonic anhydrase activity (Sauze et al., 2017) we developed a qPCR approach that helped us show that the putative natural abundance of soil phototrophs derived from the number of 23S reads were relatively small under darkened conditions compared to the bacterial (16S) and fungal (18S) abundances but their relative abundances increased significantly when incubated in the light. This probably and unsurprisingly suggests that such an influence might be somewhat dependent on the canopy cover and light conditions of the system in question.

Muriel Bristol Roach, B. (1927). On the algae of some normal English soils. *The Journal of Agricultural Science, 17*(4), 563-588. doi:10.1017/S0021859600018839

Seppey, C. V. W., Singer, D., Dumack, K., Fournier, B., Belbahri, L., Mitchell, E. A. D. and Lara, E.: Distribution patterns of soil microbial eukaryotes suggests widespread algivory by phagotrophic protists as an alternative pathway for nutrient cycling, Soil Biology and Biochemistry, 112, 68–76, doi:10.1016/j.soilbio.2017.05.002, 2017.

Sauze, J., Ogée, J., Maron, P.-A., Crouzet, O., Nowak, V., Wohl, S., Kaisermann, A., Jones, S. P. and Wingate, L.: The interaction of soil phototrophs and fungi with pH and their impact on soil CO2, CO18O and OCS exchange, Soil Biology and Biochemistry, 115(Supplement C), 371–382, doi:10.1016/j.soilbio.2017.09.009, 2017.

Wingate L., Ogée J., Cuntz M., B. Genty, I. Reiter, U. Seibt, D. Yakir, K. Maseyk , E.G. Pendall, M.M. Barbour, B. Mortazavi, R. Burlett, P. Peylin, J. Miller, M. Mencuccini, J.H. Shim, J. Hunt, J. Grace (2009) The impact of soil microorganisms on the global budget of $\delta^{18}O$ in atmospheric CO₂. *Proceedings of the National Academy of Sciences of America,* 106, 22411–22415.

11) P4L99: Be specific about "the inorganic nitrogen chemistry of soil solutions."

We have re-written the introduction in response to the comments about writing style. This section has changed accordingly.

L100 – L107: "Various anions may also play a role in controlling the activity of CAs (Tibell et al., 1984). In particular, nitrate ($NO_3^-$) has been shown to inhibit different CAs in a range of microbes and plants (Amoroso et al., 2005; Innocenti et al., 2004; Peltier et al., 1995). This suggests that variations in soil nutrient availability between ecosystems could give rise to differences in $k_{iso}$. Furthermore, the addition of common fertilisers such as ammonium nitrate ($NH_4NO_3$) to agricultural soils could have an inhibitory role on CA activity in addition to causing shifts in the size and composition of microbial communities present Indeed, this hypothesis is supported by recent $NH_4NO_3$ fertilising experiments that demonstrated decreases in the CA catalysed hydrolysis of carbonyl sulphide (Kaisermann et al., 2018b). So far, the impact of nitrates on $k_{iso}$ has not been investigated in soils."

12) P5L133–134: Does sieving affect carbonic anhydrase activity in soils?

Our experiments did not test for the impact of sieving on soil carbonic anhydrase activity and as far as we are aware this has not been reported in the literature, thus the nature of these effects is not well understood and is discussed in L366 to L388.

13) P7L195–198: What was the precision of the IRIS for CO2 and δ18O-CO2 measurements when averaged in 40 intervals?

We have added this information..

L200 - L201: "The associated precision for the total concentration and $\delta^{18}O$ of CO₂ was 0.02 ppm and 0.06 ‰ VPDBg respectively."

14) P7L210: Eq. (1) requires a steady-state condition. What is the turnover time for gas exchange in the cuvette? Could you show that the measurement period (12, P1L191) is much longer than this turnover time?

The turnover time was less than 10 minutes. We have added this information to the text. Each jar was flushed for 20 or 22 minutes before the measurement period (L193 - L194) and 22 or 24 minutes before the first measurement of the

chamber line was made. These timings reflect the need to balance the trade-off between approximate steady-state conditions and changes in the isotopic composition of the soil water pool (Jones et al. 2017).

L194: *"The turnover time of air in the jar was less than 10 minutes."*

Jones, S. P., Ogée, J., Sauze, J., Wohl, S., Saavedra, N., Fernández-Prado, N., Maire, J., Launois, T., Bosc, A. and Wingate, L. (2017) Non-destructive estimates of soil carbonic anhydrase activity and associated soil water oxygen isotope composition, Hydrology and Earth System Sciences, 21(12), 6363–6377, doi:https://doi.org/10.5194/hess-21-6363-2017.

15) P8L238–239: Please considering providing a table of site information and soil characteristics, either as a supplementary table or a metadata file in the online data set associated with this study. Although such information is available for European sites in Kaisermann et al. (2018) ACP, it would not be convenient for C4 the reader to reference across multiple publications. For the Australian sites, I do not see any such data.

We have archived the data with PANGAEA.

L662 – L664: *"The data produced in this study have been achived with PANGAEA (https://doi.org/10.1594/PANGAEA.928394). The data may also be requested from the corresponding author by email."*

16) P10L281: What does the "two-term model" mean? What are the predictors?

Two-term models are those limited to 2 or less predictive terms. We have rephrased this to make it clearer.

L286 – L 289: *"The same approach was also applied to the 27 soils from the EUR sampling campaign and extended to consider the relationships with soil texture and carbon and nitrogen contents to investigate their utility in upscaling efforts. To prevent over-fitting, these models were limited to a maximum of two of predictive terms. The predictive terms considered were soil sand, silt, clay, carbon and nitrogen content, the ratio of carbon to nitrogen content and soil pH."*

17) P10L282: Have soil texture, carbon content, and nitrogen content been considered in the aforementioned model selection procedures?

The same model selection procedures were used and this now explicitly stated in the text (L286).

18) P11L305: "Correlations between all other variable pairings were weaker and non-significant ($p > 0.05$)." - I find this observation in apparent conflict with the interpretation of NH4NO3 treatment results. If NO3- concentration does not control kiso in natural soils, why would adding NH4NO3 cause kiso to decrease through carbonic anhydrase inhibition? One possible scenario could be that the variation in kiso that is attributable to soil pH is so large that any influence from NO3- concentration is obscured. To test whether this would be the case, Spearman's rank correlation would be insufficient. You would need to control for the variation due to pH before testing the effect of [NO3-].

Spearman's rank correlation is used to identify the strongest patterns between pairs of variables without making a priori assumptions about the data. This is particularly useful as it helps us identify potential co-correlations such as that between pH and ammonium availability that may confound the subsequent analyses discussed in the paragraph following that referred to in this comment.

Subsequent use of multiple generalised linear models lets us test these relationships in a more satisfactory fashion. This analysis bears out the main result of the Spearman's rank correlation i.e. that most of the variability in $k_{iso}$ is explained by soil pH. However, after controlling for the effect of pH the inclusion of nitrate availability and biomass both significantly increase the degree of variability explained (see also Table S1). This indicates that nitrate concentration does indeed control $k_{iso}$ in natural soils. Figure 3 b shows the nature of this relationship with nitrate concentration, particularly under acidic conditions, causing $k_{iso}$ to decrease.

19) P13L357: While the fraction of explained deviance is high, this is a small sample with n=14 and uncertainty associated with # the model could be large. What is the confidence interval of the coefficient of ln NO3-?

We agree that the sample size is small and report this model simply as the best fit to the data out of the variables considered in order to understand the influence of the treatment on the rate of exchange. Indeed, the uncertainty is large particularly at higher values of change. Please see the confidence interval provided in Figure 5.

20) P13L376–380: "Whether the potential [. . . ] remains an unresolved but key question." - Not sure what you are trying to mean with this sentence. Please clarify it.

This section has been edited in response to comments about the writing style.

L380 – 388: *"Understanding why $k_{iso}$ can be orders of magnitude greater in the field compared to values observed in laboratory incubations is a key question for further studies. Potentially, the abundance and activity of CAs may be reduced during the process of sieving soils and incubating them for prolonged periods in the dark. For example, the exclusion of intact roots and mycorrhizal fungi interacting within the rhizosphere might reduce $k_{iso}$ (Li et al., 2005). Equally the suppression of phototrophic community members by incubating soils in the dark (Sauze et al., 2017) may also contribute to differences in $k_{iso}$ between the field and such experiments. Furthermore, we cannot exclude the possibility that determining $k_{iso}$ accurately under field conditions is less reliable. For example, the calculation of $k_{iso}$ relies on determining the $\delta^{18}O$ of the soil water pool in equilibrium with $CO_2$. Given the potential for increased heterogeneity in the soil water pool in natural conditions this may make it more challenging to determine $k_{iso}$ robustly in the field (Jones et al., 2017)."*

21) P15L425: "The absence of strong patterns with climate or land-cover in this study may well reflect the fact that the temperature and moisture conditions used are unrepresentative of field conditions especially for colder and drier sites." - Or, it could also be that soil texture and composition are the main controls.

It is true that the conditions experienced by the microbes in their natural environments can be very different from those experienced in our experiment. This would definitely be interesting to look at in the future with a different experimental and mechanistic modelling approach. However, the aim of the present study was to standardise moisture and temperature conditions to the best of our abilities and investigate how the gas exchange rates and enzyme activity of these different communities compared. Opting for this experimental design meant we were not able to attribute statistically whether differences in activity were underpinned by land-use or climate class in a way that would facilitate a simple scaling up approach, Our study indicates other soil traits such as pH have the potential to provide more reliable spatial predictions of $k_{iso}$. With larger databases perhaps land-use or climate patterns will begin to emerge as important large-scale drivers of soil function and predictors of soil-atmosphere gas exchange but for the moment it remains unclear as these datasets are rare in the community.

22) P15L435: What are the "pedotransfer functions?"
Pedotransfer functions are predictive functions used to estimate certain soil properties from more readily available data. We have altered this section in response to comments about the writing style and EC1 comment 26)

L445 – L460: *"A significant challenge to using this statistical relationship to predict $k_{iso}$ is underpinned by our capacity to describe the spatial and temporal variations in the important drivers of $k_{iso}$, namely soil pH, microbial biomass and exchangeable $NO_3^-$. For this reason we also considered whether more readily available parameters such as soil texture, carbon content and nitrogen content might provide an alternative basis for empirical predictions of $k_{iso}$ (Van Looy et al., 2017). However, relationships between these variables and $k_{iso}$ were relatively weak and could only explain a marginal amount of the observed variability. Fortunately, a number of promising spatial databases are evolving for soil characteristics such as pH and microbial biomass (Serna-Chavez et al., 2013; Slesserev et al., 2016). Likewise a number of land surface models can now estimate the spatial and temporal dynamics of the biosphere nitrogen cycle convincingly (Zaehle, 2013). Predictions of soil nutrient dynamics will likely depend on the use of such advanced soil nitrogen cycle models. Given the interaction between soil pH and exchangeable $NO_3^-$ (Fig. 3 a & b), the absence of such data may not seriously compromise predictions for fertilised agricultural soils as typically they are not strongly acidic. However, accurately predicting natural spatial and seasonal variability and the influence of future changes in atmospheric $NO_3^-$ deposition (DeForest et al., 2004) may be more problematic. Nonetheless, the data reported in this study now lay the foundations for an empirical approach to predicting $k_{iso}$ for a wide range of soils using readily available maps of key soil traits. This represents an important breakthrough in predicting how variations in soil community CA activity impacts the $\delta^{18}O$ of atmospheric $CO_2$."*

Technical comments

23) P1L10: "gross primary production" vs. P1L25 "gross primary productivity " (emphases mine), pick one.

"gross primary production" is used through-out the text.

24) P1L11: "ecosystem-scale" → "ecosystem scale"

This was removed as part of the improvements to the writing requested in other comments.

25) P1L15: Add a comma before "indicating [. . . ]."

A comma was added.

L21 – L 22: *"Observed values for $k_{iso}$ always exceeded theoretically-derived uncatalysed rates, indicating a significant influence of CAs on the variability of $k_{iso}$ across the soils studied."*

26) P1L33: "the leaves of plants" → "leaves". Pleonasm.

This was removed as part of the improvements to the writing requested in other comments.

27) P2L35: "causing CO2 that interacts with a leaf but is not fixed to inherit the isotopic composition of the leaf water pool" - A difficult sentence. Please clarify.

This was removed as part of the improvements to the writing requested in other comments.

28) P2L44–P3L73: This paragraph has a lot to unpack. In my opinion, to bring clarity to this paragraph, you may consider splitting it into two. Describe the abiotic reaction of oxygen isotope exchange first, and then introduce the role of carbonic anhydrases in accelerating the reaction towards equilibrium. I would consider splitting the paragraph at line 62 and rearraging sentences for a clean separation.

The introduction has been re-written in response to this and other comments about the writing style (L35 – L100).

29) P3L83: "it's" → "its"

Removed as part of the re-write of the introduction in response to comments about the writing style.

30) P3L87–89: "Such an observation may result from changes in size or composition of the microbial communities involved as discussed (Sauze et al., 2017, 2018)." - This is a reiteration of P3L79–81.

Removed as part of the re-write of the introduction in response to comments about the writing style.

31) P4L95: "non-carbon" → "non-carbonate"

This was changed as part of the response to comments about the writing style.

L93: *"Various anions may also play a role in controlling the activity of CAs (Tibell et al., 1984)."*

32) P5L123: "principle" → "principal"

Corrected (L118)

33) P5L124: "indicted" → "indicated"

Corrected (L126)

34) P6L171: This should be section 2.2, not 2.1.

Corrected (L176).

35) P11L312–316 and P12L330–337: It is inconvenient to track which model is which. Please consider listing model diagnostics in supplementary tables.

We have added three tables to the supplement listing the relevant models discussed in the text.

Table S1: Ranking and included terms for a subset of the generalised linear models tested to predict variations in the rate of oxygen isotope exchange, $k_{iso}$, for the entire dataset (n = 44). Model selection was limited to a maximum of four predictive terms and the intercept. The terms MB, $NO_3^-$ and $NH_4^+$ are the natural logarithms of microbial biomass and nitrate and ammonium availability. Selected terms or interactions within each model are indicated by + symbols whilst - symbols indicate their omission. The interactions Campaign:pH and Campaign:MB are omitted from the table for

brevity as they were not selected in any of the models shown. Model ranking was based on comparison of sample size corrected Aikake's Information Criterion (AICc) with ΔAICc indicating the difference in AICc from the best model. ΔAICc of 2 or more indicates real differences in model performance.

| Rank | Intercept | Campaign | pH | MB | NO$_3^-$ | NH$_4^+$ | Campaign: NO$_3^-$ | pH: MB | pH: NO$_3^-$ | MB: NO$_3^-$ | NO3$^-$: NH$_4^+$ | ΔAICc |
|---|---|---|---|---|---|---|---|---|---|---|---|---|
| 1 | + | - | + | + | + | - | - | - | + | - | - | 0.00 |
| 2 | + | - | + | + | + | - | - | - | - | - | - | 6.10 |
| 3 | + | + | + | + | + | - | - | - | - | - | - | 7.06 |
| 4 | + | - | + | + | + | - | - | + | - | - | - | 7.07 |
| 5 | + | + | + | - | + | - | + | - | - | - | - | 7.09 |
| 6 | + | - | + | + | + | - | - | - | - | + | - | 8.79 |
| 7 | + | + | + | - | + | - | - | - | - | - | - | 12.43 |
| 8 | + | - | - | + | + | + | - | - | - | - | + | 13.27 |
| 16 | + | - | + | - | - | - | - | - | - | - | - | 21.56 |
| 19 | + | - | - | - | - | + | - | - | - | - | - | 26.48 |
| 21 | + | - | - | + | - | - | - | - | - | - | - | 43.64 |
| 28 | + | - | - | - | - | - | - | - | - | - | - | 47.91 |
| 33 | + | + | - | - | - | - | - | - | - | - | - | 50.15 |
| 34 | + | - | - | - | + | - | - | - | - | - | - | 50.21 |

Table S2: Ranking and included terms for a subset of the generalised linear models tested to predict variations in the rate of oxygen isotope exchange, $k_{iso}$, for the relatively invariant soil properties of the EUR campaign dataset (n = 27). Model selection was limited to a maximum of two predictive terms and the intercept. The terms C, N and CN are soil carbon and nitrogen content and their ratio. Selected terms within each model are indicated by + symbols whilst - symbols indicate their omission. Model ranking was based on comparison of sample size corrected Aikake's Information Criterion (AICc) with ΔAICc indicating the difference in AICc from the best model. ΔAICc of 2 or more indicates real differences in model performance.

| Rank | Intercept | pH | Sand | Silt | Clay | C | N | CN | ΔAICc |
|---|---|---|---|---|---|---|---|---|---|
| 1 | + | + | - | - | + | - | - | - | 0.00 |
| 2 | + | + | + | - | - | - | - | - | 0.57 |
| 3 | + | + | - | - | - | + | - | - | 1.32 |
| 4 | + | + | - | - | - | - | - | - | 1.85 |
| 5 | + | + | - | - | - | - | - | + | 1.92 |
| 6 | + | + | - | + | - | - | - | - | 2.46 |
| 7 | + | + | - | - | - | - | + | - | 4.57 |
| 8 | + | - | - | - | - | - | + | - | 21.26 |
| 9 | + | - | - | - | - | - | - | - | 22.07 |

Table S3: Ranking and included terms for a subset of the generalised linear models tested to predict variations in the change in rate of oxygen isotope exchange, $k_{iso}$, following ammonium nitrate addition (n = 15). Model selection was limited to a maximum of one predictive term and the intercept. The terms MB, NO$_3^-$ and NH$_4^+$ are differences in microbial biomass and nitrate and ammonium availability following ammonium nitrate addition whilst the prefix ln indicates the natural logarithm of these differences. Selected terms within each model are indicated by + symbols whilst - symbols indicate their omission. Model ranking was based on comparison of sample size corrected Aikake's Information Criterion (AICc) with ΔAICc indicating the difference in AICc from the best model. ΔAICc of 2 or more indicates real differences in model performance.

| Rank | Intercept | Campaign | pH | MB | NO₃⁻ | NH₄⁺ | lnMB | lnNO₃⁻ | lnNH₄⁺ | ΔAICc |
|------|-----------|----------|----|----|------|------|------|--------|--------|-------|
| 1 | + | - | - | - | - | - | - | + | - | 0.00 |
| 2 | + | - | - | - | + | - | - | - | - | 8.65 |
| 3 | + | - | + | - | - | - | - | - | - | 13.20 |
| 4 | + | - | - | - | - | - | - | - | - | 15.95 |
| 5 | + | + | - | - | - | - | - | - | - | 17.38 |
| 6 | + | - | - | - | - | - | + | - | - | 18.34 |
| 7 | + | - | - | + | - | - | - | - | - | 18.80 |
| 8 | + | - | - | - | - | - | - | - | + | 19.10 |
| 9 | + | - | - | - | - | + | - | - | - | 19.21 |

36) Figure 3: It is difficult to distingush high values from low values indicated by the color bars. Try to increase the contrast.

Figure 3 has been revised to hopefully increase the contrast of the plot gradients and use a more accessible colour palette.

[Figure]

37) Figure S1: Remove the ocean background and other unnecessary information. Please simplify this figure to make the ecoclimatic classification more evident. Consider putting the legend outside of the figure canvas to avoid interference.

Figure S1 has been revised to mask the ocean, move the legend outside of the map area and reduce the classes to only reflect those covered by the samples obtained.

[Figure]

**SC1**

1) I have read your work with great interest. The exchange of oxygen isotopes between $CO_2$ and soil water is an important process for $\delta^{18}O$, and this work contributes to a better understanding of that exchange. However, this exchange is also of great importance for the budget of $\Delta^{17}O$ in $CO_2$, a different tracer for GPP. $\Delta^{17}O$ in $CO_2$ was first proposed as a tracer of GPP by Hoag et al. (2005). More recently, laboratory studies confirmed the effect of photosynthesis on $\Delta^{17}O$ in $CO_2$ (Adnew et al., 2020), and we simulated large-scale variations of $\Delta^{17}O$ in atmospheric $CO_2$ (Koren et al., 2019). We struggled with representing the soil exchange in that model, and for follow-up studies we can possibly improve our representation of soil exchange using Eq. 6 from your manuscript. I think you can reach a greater audience if you also explicitly address the $\Delta^{17}O$ community in your work.

We have now added a couple of sentences in the introduction to clarify this point.

L35 – L45: *"Quantifying the carbon storage potential of terrestrial ecosystems and its sensitivity to climate change relies on our ability to obtain observational constraints of photosynthesis and respiration at large scales (Beer et al., 2010). Over recent decades there has been increasing interest in using the oxygen isotope composition ($\delta^{18}O$ and $\delta^{17}O$) of atmospheric carbon dioxide ($CO_2$) to trace these large and opposing $CO_2$ fluxes. This is possible because the $\delta^{18}O$ of leaf-atmosphere $CO_2$ exchange is relatively enriched in $^{18}O$ compared to that of atmospheric $CO_2$ and the $\delta^{18}O$ of soil-atmosphere $CO_2$ exchange (Francey & Tans, 1987; Wingate et al., 2009; Welp et al., 2011). Similarly, photochemical processes in the stratosphere cause anomalies between the $\delta^{17}O$ and $\delta^{18}O$ of atmospheric $CO_2$ that are subsequently reset during leaf-atmosphere $CO_2$ exchange (Hoag et al., 2005; Koren et al., 2019; Adnew et al., 2020). However, the routine use of these tracers to constrain the photosynthetic term of the atmospheric mass budget for the $\delta^{18}O$ and $\delta^{17}O$ of $CO_2$ has been hampered by an incomplete understanding of how the influence of soil-atmosphere $CO_2$ exchange varies across different soil types and environmental conditions. Here we focus on $\delta^{18}O$ but the key challenges to understanding these variations are also relevant to considerations of $\delta^{17}O$."*

2) In the first line and last line of the abstract I would replace "$\delta^{18}O$" with "$\delta^{18}O$ and $\Delta^{17}O$".

We have rephrased the abstract to remove the emphasis on only $^{18}O$ and replaced this with a more general reference to the oxygen isotope composition of atmospheric $CO_2$ (L11 - L31).

3) Sec 2. Are you sure that the sampling and transporting of soil samples does not affect the CA or microbes in the sample?

We do indeed expect there to be a disturbance effect on the microbial community when transporting soils and sieving them, thus it is important to be mindful of this when comparing results from soils measured under field conditions and

those measured in laboratory experiments as well as extrapolating results from mesocosms to the large scale. This study however was designed to characterize a set of homogenized climate-controlled soils to make a link between the measured CA activity, the mesocosm soil characteristics and their response to changes in inorganic N concentrations. However the quantitative influence of transport and sieving on carbonic anhydrase activity is so far not well understood but is discussed. Please see L366- L388 in the Discussion.

4) There are two sections with number 2.1.

Corrected (L176).

5) L139: "Tillburg". This should be the lovely city "Tilburg".

Corrected (L142).

6) L147: Why did you choose to report on the VPDBg scale, instead of e.g. VSMOW?

We preferentially report our $CO_2$ in air measurements on the VPDBg scale (also known as VPDB-$CO_2$ scale) reflecting the fact that values assigned to our working standards are ultimately tied to the acid digestion of RM NBS-19 calcite. Please see:

Werner, R. A., Rothe, M. and Brand, W. A.: Extraction of $CO_2$ from air samples for isotopic analysis and limits to ultra high precision $\delta^{18}O$ determination in $CO_2$ gas, Rapid Communications in Mass Spectrometry, 15(22), 2152–2167, doi:https://doi.org/10.1002/rcm.487, 2001.

Werner, R. A. and Brand, W. A.: Referencing strategies and techniques in stable isotope ratio analysis, Rapid Communications in Mass Spectrometry, 15(7), 501–519, doi:https://doi.org/10.1002/rcm.258, 2001.

7) L210: The units provided in the text do not agree with Eq. 1.

Corrected.
L215: "*where u is the flow rate (mol s$^{-1}$) through the chamber line*".

8) L423: I would briefly mention Δ17O here.

We now reference this in the discussion.
L435: "*Improvements in our ability to predict soil k iso and its influence on the δ 18 O of atmospheric CO 2 are important in refining the use of this tracer and others such as 17 O to constrain photosynthesis and respiration at large scales (Wingate et al., 2009; Welp et al., 2011; Koren et al., 2020).*"

9) Caption Fig. 1: The authors mention twice: "dissolved organic carbon (DIC)". Should this be DIC or DOC?

Corrected to DIC.